# The molecular structure of IFT-A and IFT-B in anterograde intraflagellar transport trains

Samuel E. Lacey, Helen E. Foster & Gaia Pigino ✉

Anterograde intraflagellar transport (IFT) trains are essential for cilia assembly and maintenance. These trains are formed of 22 IFT-A and IFT-B proteins that link structural and signaling cargos to microtubule motors for import into cilia. It remains unknown how the IFT-A/-B proteins are arranged into complexes and how these complexes polymerize into functional trains. Here we use in situ cryo-electron tomography of *Chlamydomonas reinhardtii* cilia and AlphaFold2 protein structure predictions to generate a molecular model of the entire anterograde train. We show how the conformations of both IFT-A and IFT-B are dependent on lateral interactions with neighboring repeats, suggesting that polymerization is required to cooperatively stabilize the complexes. Following three-dimensional classification, we reveal how IFT-B extends two flexible tethers to maintain a connection with IFT-A that can withstand the mechanical stresses present in actively beating cilia. Overall, our findings provide a framework for understanding the fundamental processes that govern cilia assembly.

Cilia are hair-like organelles that extend from eukaryotic cells and beat to create motion (motile cilia) or act as a hub for signaling (primary cilia). At their core is a ring of nine interconnected microtubule doublets in a structure known as the axoneme (Fig. 1a). A diffusion barrier exists at the base of the cilium, meaning that the vast quantities of structural proteins required to build the axoneme need to be delivered by microtubule motors in a process called intraflagellar transport (IFT). IFT also transports membrane-associated proteins into and out of the cilium to regulate key developmental signaling pathways[1]. Underlining the importance of IFT, the absence of many IFT proteins is lethal and mutations leading to variations of IFT-related proteins can result in a group of congenital diseases called ciliopathies, with diverse phenotypes[2].

IFT is organized by the IFT-A and IFT-B protein complexes. Together, these assemble into ordered and repetitive IFT trains that link the microtubule motors to IFT cargos. The IFT process is initiated at the base of the cilium, where IFT-B complexes start to polymerize on their own[3]. This nascent train acts as a platform for IFT-A polymerization and recruits kinesin-2 motors (Fig. 1a). The structural and signaling cargos then dock to the train, as well as autoinhibited cytoplasmic dynein-2 motors. Kinesin carries the train into the cilium and delivers the train and its cargos to the tip[4,5]. The IFT-A/-B components then remodel into a conformationally distinct retrograde train, which

rebinds to the now active dynein-2 and transports a new selection of cargos back to the cell body[6–8].

From our previous cryo-electron tomography (cryo-ET) study of in situ *Chlamydomonas reinhardtii* cilia, we know the overall appearance of anterograde trains to 33–37 Å resolution[9]. IFT-B, which contains 16 proteins (IFT172, 88, 81, 80, 74, 70, 57, 56, 54, 52, 46, 38, 27, 25, 22 and 20), forms a 6-nm repeat with one autoinhibited dynein-2 bound every third repeat (Fig. 1b). IFT-A, which contains six proteins (IFT144, 140, 139, 122, 121 and 43), sits between IFT-B and the membrane. It has an 11.5-nm repeat, creating a mismatch in periodicity between IFT-A and IFT-B. However, due to the limited resolution, the molecular architectures of IFT-A and IFT-B remain unknown. Crystal structures of some IFT-B proteins have been solved[10–15], but they are mostly of small fractions of the overall proteins. Much of our knowledge therefore comes from biochemically mapped interactions between isolated IFT-B proteins[10,11,16]. None of the six IFT-A components have been structurally characterized and there are fewer verified interactions for this complex[16–18].

As a result, we have a limited understanding of many fundamental mechanisms underlying IFT. To address this, we generated substantially improved (10–18 Å) subtomogram averages of *Chlamydomonas* IFT trains, allowing us to build a complete molecular model of the

Human Technopole, Milan, Italy. ✉e-mail: gaia.pigino@fht.org

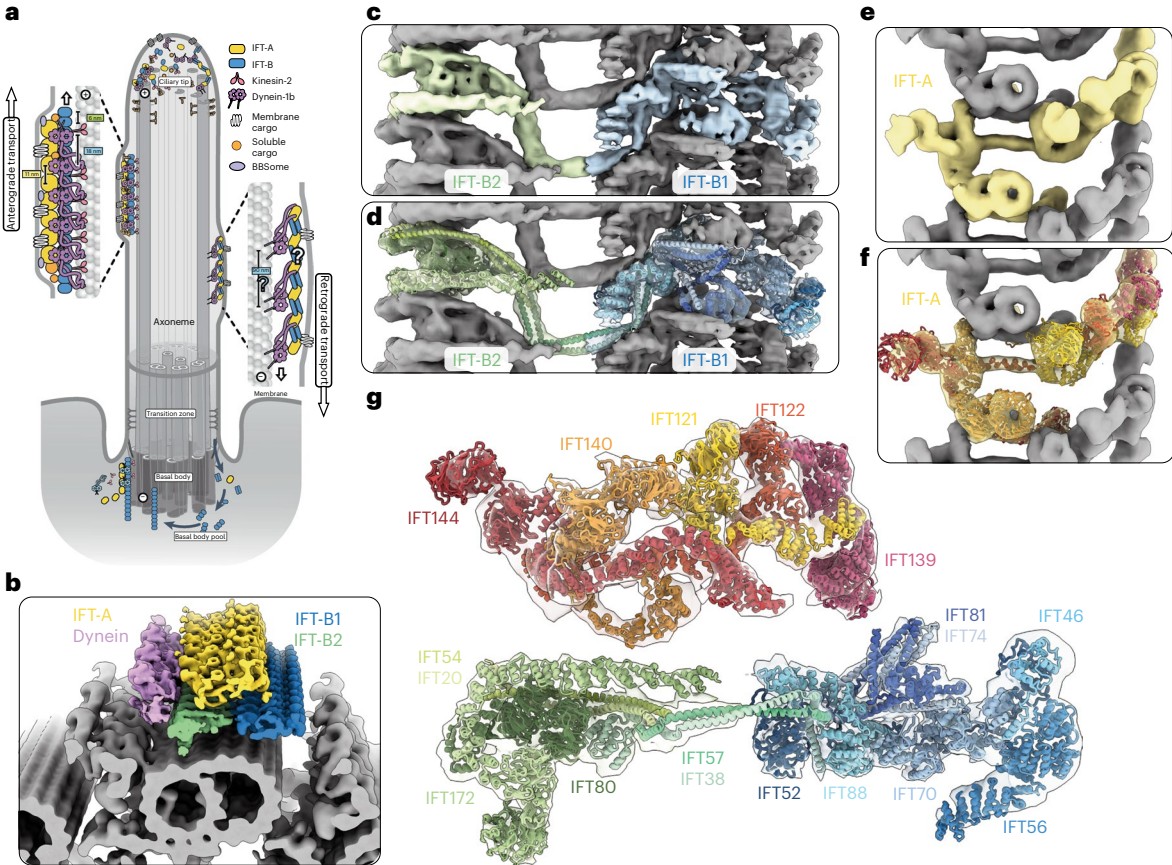

**Fig. 1 | An overview of the anterograde IFT train structure. a**, Cartoon model of IFT within a cilium. Anterograde trains form at the base of the cilium (basal body) and carry cargo through the diffusion barrier (transition zone) and to the tip. Here, they remodel into retrograde trains that carry their cargos back to the basal body for recycling. **b**, The new subtomogram averages lowpass filtered and colored by complex (yellow, IFT-A; blue, IFT-B1; green, IFT-B2; purple, dynein), docked onto a cryo-ET average of the microtubule doublets found in motile cilia. One repeating unit is highlighted in each complex with darker shading. **c**, The new subtomogram averages for IFT-B1 (blue) and IFT-B2 (green), displayed together as a composite. One repeating unit is highlighted in color, with the adjacent repeats in gray. **d**, Equivalent to **c**, but with the highlighted repeat now shown partially transparent and our molecular model of IFT-B docked in. **e**, The new subtomogram average of IFT-A, with one repeating unit shown in yellow and adjacent repeats in gray. **f**, Equivalent to **e**, but with the highlighted repeat now shown partially transparent and our molecular model of IFT-A docked in. **g**, Our molecular model of one repeating unit of IFT-A and IFT-B in the anterograde train, shown in cross-section as if looking down the microtubule. The partially transparent density for four maps is shown: IFT-B2 and IFT-A, with the main IFT-B1 average combined with a masked refinement of the region containing IFT56 (IFT-B1 tail; Extended Data Fig. 2a), since this region is more flexible relative to the core.

anterograde train. Here, we present a tour of the IFT-A and IFT-B complexes within the context of polymerized trains. Together, our results provide insights into the organization and assembly of IFT trains, how cargos are bound to the train and the conversion of anterograde trains into retrograde trains.

## Creating a model of anterograde IFT trains

To generate a molecular model of the anterograde IFT train, we collected 600 cryo-electron tomograms of *Chlamydomonas* cilia. We picked and refined IFT-B and IFT-A repeats independently due to their periodicity mismatch[9] and performed subtomogram averaging with the STOPGAP–Warp/M–Relion 3 processing pipeline (Extended Data Figs. 1–3). In IFT-B, we identified two rigid bodies that flex around a central hinge that correspond to the biochemically characterized IFT-B1 and IFT-B2 subcomplexes (Extended Data Fig. 2a). After masked refinements, we obtained structures at 9.9 Å resolution for IFT-B1, 11.5 Å resolution for IFT-B2 and 18.6 Å resolution for IFT-A (Fig. 1c,e, Extended Data Figs. 2e,f and 3g,h and Table 1).

To understand how the IFT proteins are organized in their complexes, we built a molecular model into our maps. As de novo model building is not possible at this resolution, we used a hybrid approach

by flexibly fitting high-confidence AlphaFold2 models of IFT proteins (Table 1). This allowed us to build a molecular model of the complete anterograde train (Fig. 1d,f,g, Extended Data Figs. 4a,b and 5a–c and Supplementary Videos 1 and 2).

## IFT-B is organized around IFT52

IFT-B is central to the assembly of anterograde trains. It recruits active kinesin motors and carries both the IFT-A complex and the retrograde motor dynein-2 to the tip[19] (Fig. 1b). IFT-B is also responsible for the recruitment of all characterized structural cargos to anterograde trains. It is an elongated complex with two distinct lobes corresponding to IFT-B1 and IFT-B2 (Fig. 2a–d). Our structure reveals the crucial role that the IFT-B1 component IFT52 plays in the structural integrity of the entire IFT-B complex.

IFT52 consists of an amino (N)-terminal GIFT (GldG, intraflagellar transport) domain, a central disordered region and a carboxy (C)-terminal domain (CTD) that forms a heterodimer with IFT46 (ref. 11) (Fig. 2e and Extended Data Fig. 4a). It spans the length of IFT-B1, with the GIFT domain on the microtubule doublet-proximal surface at the center of the train and the IFT52-CTD:IFT46 heterodimer at the periphery (Fig. 2a,b). IFT88 and IFT70—two supercoiled tetratricopeptide

**Table 1 | Cryo-EM data collection, refinement and validation statistics**

|  | IFT-A average | IFT-B1 average | IFT-B2 average |
|---|---|---|---|
| Data collection and processing |  |  |  |
| Magnification | 33,000× | 33,000× | 33,000× |
| Voltage (kV) | 300 | 300 | 300 |
| Tilt range/increments (°) | ±60/3 | ±60/3 | ±60/3 |
| Electron exposure (e⁻ Å⁻²) | 100 | 100 | 100 |
| Defocus range (μm) | −3 to −4.5 | −3 to −4.5 | −3 to −4.5 |
| Pixel size (Å) | 3.03 | 3.03 | 3.03 |
| Symmetry imposed | C1 | C1 | C1 |
| Final particle images (number) | 3,897 | 18,216 | 18,216 |
| Map resolution/FSC threshold (Å) | 20.5/0.143 | 9.9/0.143 | 11.4/0.143 |
| Refinement |  |  |  |
| Map sharpening B factor (Å²) | −2,700 | −450 | −700 |
| Validation |  |  |  |
| MolProbity score | 2.41 | 2.18 | 2.18 |
| Clashscore | 23.9 | 16.7 | 16.7 |
| Poor rotamers (%) | 0.12 | 0.07 | 0.07 |
| Ramachandran plot |  |  |  |
| Favored (%) | 90.3 | 92.7 | 92.7 |
| Disallowed (%) | 0.13 | 0.1 | 0.1 |
| FSC (model to map; 0.5 threshold) | 21.4 | 10.2 | 12.1 |

The Electron Microscopy Data Bank accession codes for IFT-A, IFT-B1 and IFT-B2 are EMD-15980, EMD-15978 and EMD-15979, respectively. The Protein Data Bank codes are 8BDA, 8BD7 and 8BD7, respectively. FSC, Fourier shell coefficient.

repeat (TPR) proteins—wrap around the central disordered domain of IFT52 by stacking end to end to create a continuous central pore (Fig. 2e and Extended Data Fig. 6a,b,f). IFT70 is known to make a tight spiral with a hydrophobic core and IFT52 is thought to be an integral part of its internal structure[11]. However, we see that IFT88 forms a more open spiral with charged internal surfaces, suggesting that its interaction with IFT52 is reversible. The remainder of IFT-B1 is assembled around the IFT88/70/52 trimer, which binds to the coiled-coil IFT81/74 subcomplex and IFT56, a third TPR spiral protein (Extended Data Fig. 6d,e). Therefore, the IFT-B1 subcomplex is assembled around IFT52.

Additionally, IFT52 and IFT88 form the main interface between IFT-B1 and IFT-B2. This is mediated through interactions with IFT57/38 of IFT-B2, consistent with biochemical data[10]. IFT57/38 is a segmented coiled coil, with both proteins also containing an N-terminal calponin homology (CH) domain. IFT38-CH was previously shown to form a high-affinity interaction with the N-terminal WD40 repeat domain (WD) of IFT80 (ref. 15). In our structure, this interaction anchors IFT57/38 in IFT-B2 (Extended Data Fig. 6g). The coiled coils extend across the central region to contact IFT88 from the neighboring repeat (Fig. 2b). Here, conserved proline residues in IFT57 and IFT38 create a right-angled kink (Extended Data Fig. 6h) that points the subsequent coiled-coil segment toward the IFT88 in the same repeat. The loose spiral of IFT88 creates an open cleft, which IFT57/38 and the IFT52 disordered region slot into, creating multiple contacts between the IFT-B1 and IFT-B2 components (Fig. 2f).

Taken together, we find that IFT52 is the cornerstone of the IFT-B complex. This is consistent with results from the *Chlamydomonas bld1* mutant, which lacks functional IFT52 and cannot grow cilia or form

IFT-B complexes[20,21]. Furthermore, in humans, a mutation leading to altered IFT52 at the interface with IFT57/38 (causing substitution of aspartic acid with histidine at residue 259 of IFT52 and corresponding to the substitution of aspartic acid at residue 268 of IFT52 in *Chlamydomonas* (Extended Data Fig. 6i)) is associated with a developmental kidney ciliopathy[22], which could be caused by destabilization in the association of IFT-B1 and B2.

## IFT81/74 is stabilized by neighboring repeats

Next, we wanted to understand how the individual IFT-B1 complexes associate as polymers. Part of the interaction is mediated by simple wall-to-wall contacts between adjacent IFT88/70/52 trimers (Fig. 2b). These contacts are supplemented by a more intricate network of lateral interactions in the IFT81/74 dimer that sits on top of IFT88/70/52. IFT81/74 forms eight coiled-coil segments (CC1–8)[11,13]. The loop between IFT81/74-CC1 and -CC2 forms the main attachment to the IFT-B1 core by binding to the same cleft in IFT88 as in IFT57/38 (Fig. 2f,g). The first four coiled-coil segments then form two interactions with adjacent IFT-B1 repeats, forcing them into a folded/compressed conformation (Fig. 2h). First, the N-terminal IFT81-CH domain is raised above the IFT88/70/52 trimer through an interaction between IFT81/74-CC1 and IFT70 of the neighboring repeat. Then, IFT81-CH acts as a strut against which CC2/3 from the neighboring repeat leans in an upright position. Since the coiled-coil segments are linked by flexible loops, this suggests that a feature of IFT-B polymerization is the cooperative stabilization of IFT81/74 in a compressed conformation. Furthermore, this conformation positions the flexible C-terminal half of IFT81/74, which recruits the IFT27, IFT25 and IFT22 subunits[11,13], toward the membrane (Fig. 2a,g). This allows IFT27/25/22 to fulfill proposed roles in the recruitment of membrane cargos[23,24] and provides sufficient flexibility to maintain an interaction with proteins in the crowded ciliary membrane.

## IFT80 forms the core of IFT-B2

The IFT-B2 subcomplex forms the second lobe of IFT-B (Extended Data Fig. 7a–d and Supplementary Video 1). It is made up of two pairs of coiled-coil proteins (IFT57/38 and IFT54/20) and two large proteins (IFT172 and IFT80), which each contain a pair of tandem WD domains followed by C-terminal TPR motifs (Extended Data Fig. 4a,b). The second WD domain of both of these proteins forms an uncommon incomplete circle (Fig. 3a–c and Extended Data Fig. 7f), particularly dramatically in the case of IFT172.

From our structure, we see that IFT80 is at the center of the IFT-B2 subcomplex, with much of its surface covered by protein interactions (Fig. 3a,b). The IFT80 WD domains are sandwiched between the WD and TPR domains of two neighboring copies of IFT172 (Fig. 3a,c). Previous work suggested that IFT80 homodimerizes in the initial TPR region[15], but it is monomeric in our average. Instead, IFT80-TPR wraps around the N-terminal TPR motifs of IFT172 from the neighboring repeat. IFT172 contains an extended TPR domain that is not reinforced through the formation of a superhelical twist like IFT88/70, meaning that it is likely to be more conformationally flexible. The remaining IFT172-TPR region wraps around the edge of IFT-B2 and runs toward the center of the train, forming the roof of the complex (Fig. 2a). In summary, IFT80 organizes the core architecture of the IFT-B2 complex, as well as forming an extended lateral interface capable of stabilizing flexible domains upon polymerization.

## IFT57-CH prevents IFT172-WD1 from interacting with membranes

The IFT172-WD domains were previously shown to bind to and remodel membranes in vitro, suggesting that IFT172 may play a role in membrane trafficking[25]. However, membrane binding was mutually exclusive with an interaction between IFT57-CH and IFT172-WD. We wanted to see whether this interaction is present in active anterograde trains. In our structure, IFT172-WD1 protrudes from the periphery of IFT-B2 and is

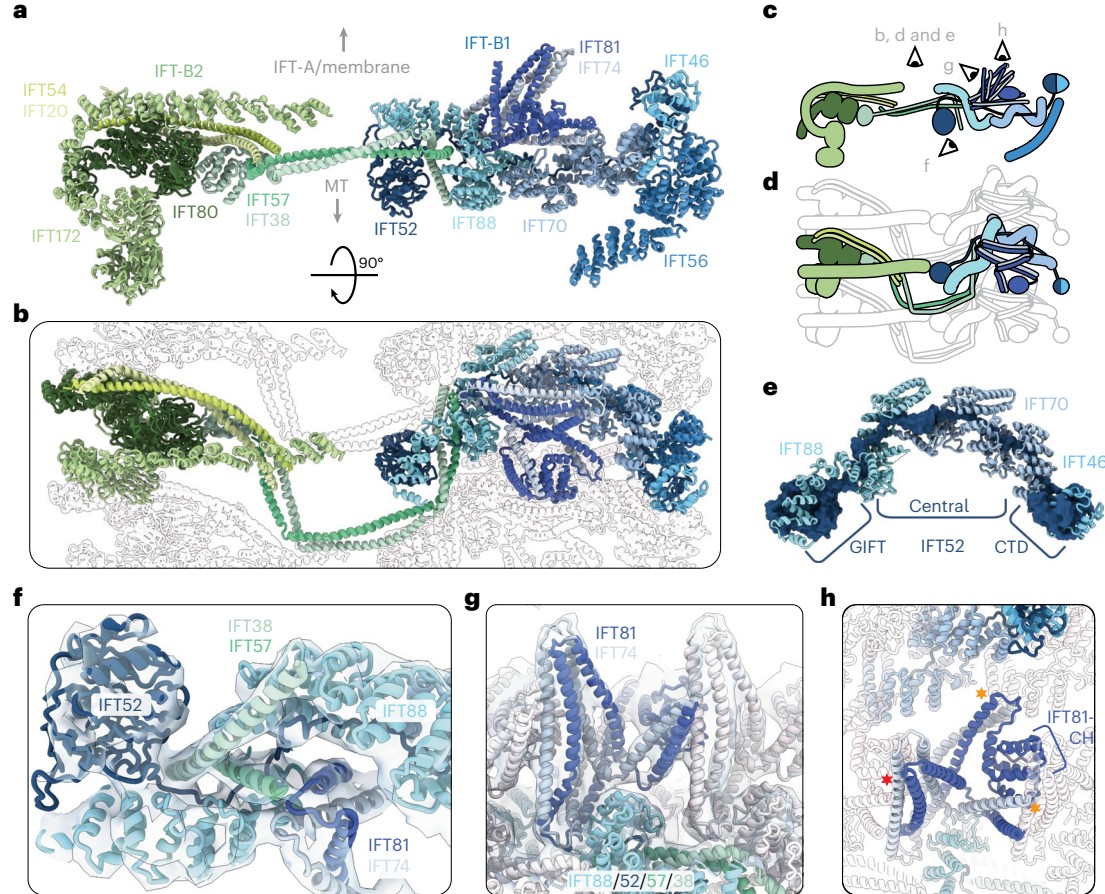

**Fig. 2 | IFT52 is central to the overall IFT-B complex. a**, One repeat of the IFT-B complex viewed in profile, looking down the train. MT, microtubule doublet. **b**, Top view of the IFT-B polymer, as if looking down from the membrane/IFT-A. A single repeat is shown in color, with adjacent repeats shown in silhouette. The coloring is as in **a**. **c**, Cartoon representation of **a**, showing the viewing positions of other panels in the figure. **d**, Cartoon representation of **b**. **e**, IFT52 (dark blue), shown as a molecular surface, forms the core of the IFT-B1 complex, with the central unstructured domain threading through the TPR superhelices of IFT88

(cyan) and IFT70 (steel blue). **f**, IFT57/38 (dark and light green, respectively) from IFT-B2 interact with IFT-B1 by fitting into a cleft in the TPR superhelix of IFT88 (cyan) along with the unstructured IFT52 central domain (dark blue). **g**, IFT81/74 (navy blue and gray, respectively) sit on top of IFT88 and form a compressed segmented coiled coil repeating along the IFT train. **h**, Top view of **g**. Lateral interactions with IFT81/74 in adjacent repeats are highlighted with stars (red star, IFT81-CH on N − 1 repeat; orange stars, IFT81/74-CC and IFT70 of N + 1 repeat).

more flexible. However, masked refinement of this region shows a clear bulge in the density that can be explained by IFT57-CH binding to IFT172-WD1 (Extended Data Fig. 7e). This interaction is possible due to the long unstructured linker between IFT57-CH and the C-terminal coiled-coil region that interacts with IFT38 (Extended Data Fig. 4a). This therefore suggests that IFT57-CH helps remove IFT172 from its putative membrane trafficking phase and makes it available for incorporation into assembling trains.

### The coiled coils in IFT-B are in a compressed conformation

Like IFT81/74 of IFT-B1, a segmented coiled coil in IFT-B2 formed by IFT57/38 is folded into a compressed conformation through lateral interactions with neighboring repeats. IFT57/38 is anchored to IFT-B2 through the IFT38-CH/IFT80 interaction (Extended Data Fig. 6g). This is supplemented by the formation of a short four-helix bundle with IFT54/20, which is a single continuous coiled coil that bridges the gap in IFT80-WD2 and runs down to the center of the train (Fig. 3a and Extended Data Fig. 7f). The helical bundle forms lateral interactions with IFT57/38 in the neighboring repeat, stabilizing a kink between segments to point it toward the IFT-B1 subcomplex (Fig. 3d). This is a second right-angle corner between IFT57/38 segments stabilized by the neighboring repeat, after the contact with IFT88 in IFT-B1 (Extended

Data Fig. 6h). We previously showed that retrograde trains have a much longer repeat than anterograde trains (~45 nm versus 11.5 or 6 nm for IFT-A and IFT-B, respectively)[9], despite being made of the same constituents[9]. We hypothesize that the compressed coiled coils in anterograde trains can be utilized during remodeling by extending into elongated conformations while maintaining intracomplex interactions.

### IFT-B cargo-binding regions face the exterior of the complex

The main role of anterograde IFT is to deliver structural and signaling cargos from the cell body to the cilium. Biochemical studies have identified several interactions between these cargos and individual IFT proteins, which we can now pinpoint to specific locations of the train. The axonemal outer and inner dynein arms are linked through their specific adapters to IFT46 and IFT56, respectively[4,26–29]. These large structural cargos will therefore be docked on the peripheral surface of IFT-B1 (Extended Data Fig. 8a). Furthermore, the N terminus of IFT70 is located on the same patch of IFT-B1 and is thought to recruit a variety of membrane proteins in humans and *Chlamydomonas*[30,31] This region of the train presents the largest open surface of IFT-B and was observed to contain heterogeneous extra densities in raw electron tomograms[9]. Therefore, we would anticipate that other large structural cargos would be engaged in similar interactions with the same IFT proteins.

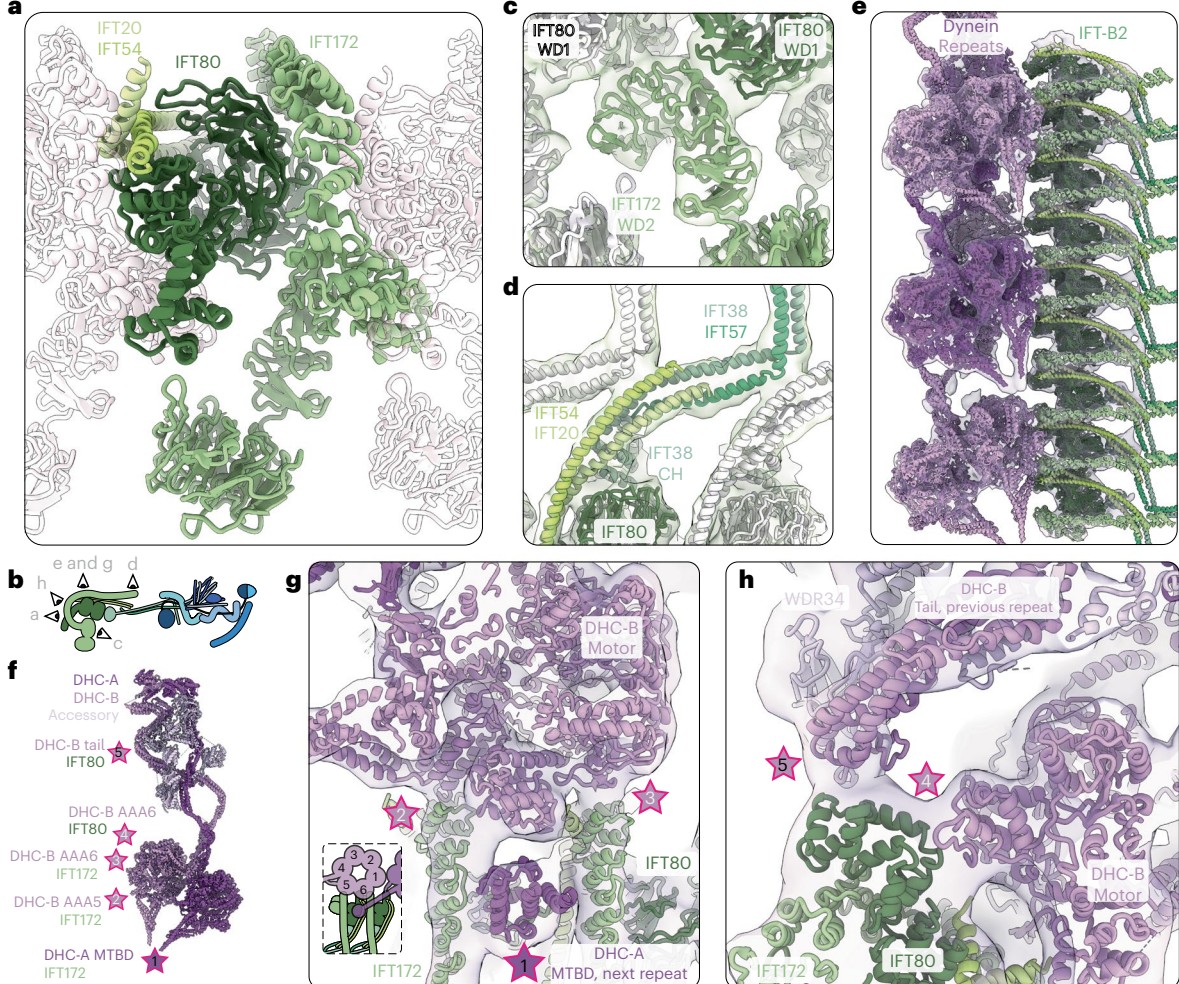

**Fig. 3 | Interaction between IFT-B2 and dynein-2. a**, IFT80 (dark green) forms the core of the IFT-B2 complex. It is surrounded by IFT172 (olive green) and the IFT54/20 (lime green and pale green, respectively) coiled coil. Adjacent repeats are shown in silhouette. **b**, Cartoon representation of IFT-B depicting the positions of the views in the other panels. **c**, The second WD domain of IFT172 (olive green) does not close into a ring, and bridges two IFT80 subunits (dark green from the same complex and white in the neighbor). **d**, In the center of the complex, IFT54/20 (lime and pale green, respectively) and IFT57/38 (turquoise

and mint green, respectively) coiled coils stack on top of each other, stabilizing a kink in IFT57/38 to point the subsequent coiled coils toward IFT-B1. **e**, The flexibly refined dynein models (purple and pink) docked into the 16 Å dynein density, along with the IFT-B2 model. **f**, Cartoon representation of cytoplasmic dynein-2 refined into our density, with the points that contact IFT-B2 and the protein they interact with highlighted with stars. **g**, Top view of the train, showing the first three contact points between dynein and IFT-B2. **h**, The two remaining contact points between dynein and the edge of IFT-B2, at the C terminus of IFT80.

Soluble tubulin is an IFT cargo thought to be recruited by a tubulin-binding module composed of IFT81-CH and the basic N terminus of IFT74 (refs. 14,32). In our structure, the residues in IFT81-CH that are important for tubulin binding lie in a narrow gap between coils that prevents an interaction (Extended Data Fig. 8b). Alternatively, IFT81-CH could bind to tubulin in the same way as the structurally conserved CH domain of kinetochore protein Ndc80 (ref. 33) (Extended Data Fig. 8c). However, this would lead to strong steric clashes with IFT81/74 in neighboring repeats (Extended Data Fig. 8d). This leaves the possibility that the IFT81/74 module binds to the acidic and unstructured C termini of tubulin, although this would be an unusual way for a CH domain to bind tubulin.

## Cytoplasmic dynein-2 interfaces require IFT-B polymerization

The retrograde IFT motor dynein-2 is transported as a cargo of antegrade trains to the tip of cilia, where it is used to transport retrograde trains back to the cell body. Previously, we showed that autoinhibited dynein-2 complexes dock onto IFT-B in a regular repeat, on the edge of

what we now determine to be IFT-B2 (ref. 9). We wanted to understand the molecular basis for this recruitment; however, the dynein density was averaged out of our overall structure since its repeat is three times that of IFT-B. To address this, we used three-dimensional (3D) classification to find dyneins in the same register. We then performed local refinements on this subclass to obtain an improved 16.6 Å final map of dynein-2, and flexibly fit the single-particle structure of human dynein-2 (ref. 34) into it (Fig. 3e and Extended Data Fig. 7g–i).

The dynein dimer consists of two dynein heavy chains (DHC-A/-B) that are split into an N-terminal tail domain and a C-terminal AAA+ motor domain[34]. The tail is used for dimerization and recruitment of accessory chains, and the motor domain generates force and binds to microtubules through a microtubule-binding domain (MTBD).

Dynein-2 binds to IFT-B2 at five contact points (Fig. 3f–h). The first is a composite surface between two IFT-B2 complexes that is only formed upon polymerization. Here, the MTBD of DHC-A sits in a trench formed between two neighboring IFT172-TPRs, with IFT80-WD2 and IFT54/20 forming the base. This interaction could be mediated by a negatively charged patch on IFT80-WD2, mimicking the interaction

between the MTBD and the negatively charged microtubule surface (Extended Data Fig. 7l,m). Two more contacts are made by the motor domain of DHC-B bridging the same two IFT172 subunits through the AAA5/6 domains. The DHC-B AAA6 domain makes an additional contact with IFT80-TPR (Fig. 3f–h). Finally, the tail of DHC-B from the adjacent dynein repeat contacts the same region of the IFT80-TPR. These contacts could be supplemented by additional, unstructured contacts like the reported interaction between the disordered N terminus of IFT54 and dynein[35].

Therefore, we find that dynein-2 is only able to bind to IFT-B2 in the context of an assembled anterograde train. Its binding site includes the TPR domain of IFT172, which is stabilized in trains but is likely to be flexible in solution based on the AlphaFold2 ensemble confidence predictions (Extended Data Fig. 4b). This, combined with the MTBD binding site that sits on the boundary between IFT-B repeats, means that dynein will only be able to form weak interactions with unpolymerized IFT-B. This provides a level of regulation to prevent dynein-2 from binding to individual IFT-B components before train assembly.

## The IFT-A polymer is continuously interconnected

The IFT-A complex sits between the IFT-B complex and the membrane (Fig. 1b). In anterograde trains, it is responsible for transport of some membrane cargos. IFT-A is made up of five structural proteins (IFT144, 140, 139, 122 and 121) and one disordered protein (IFT43). IFT144, IFT140, IFT122 and IFT121 all have tandem N-terminal WD domains followed by extended TPR domains (Extended Data Fig. 4a). IFT139 consists solely of TPR repeats, which were predicted by AlphaFold2 to form a superhelical spiral. However, how these proteins are organized into the IFT-A complex, and how the complexes assemble into polymers, could not be resolved in previous studies.

The resolution of our IFT-A reconstructions was limited to 18.6 Å (Extended Data Fig. 3g,h), potentially making subunit placement difficult. However, the AlphaFold2 models of each of the four WD-containing IFT-A proteins showed unique combinations of angles between the two WD domains and the position of the first TPR repeat (Extended Data Fig. 9a,b). This allowed us to unambiguously place the WD domains in our map and fit the C-terminal TPR domains into the connected continuous tubular densities (Extended Data Fig. 9c,d). Finally, we identified a spiral density corresponding to IFT139 to complete our model (Supplementary Video 2).

We also see an extra density at lower thresholds bridging the gap between IFT144-WD and IFT140-WD (Extended Data Fig. 9e). We do not locate the disordered IFT43 in our overall model. However, since IFT43 is thought to interact with two proteins (IFT121 and IFT139; refs. 16,18) that we show are at the other end of the complex, it is unlikely that this density corresponds to IFT43. Therefore, the density belongs to another, unidentified protein.

Our model shows that IFT-A is an intricately interconnected complex. IFT144-WD defines one end of the IFT-A complex (Fig. 4a–c) and projects out toward the membrane. The IFT140-WD domains are nearby and the N-terminal TPR motifs of IFT144 and IFT140 have a long interface running along the edge of the complex (Fig. 4b). Surprisingly, IFT144-TPR and IFT140-TPR run into the neighboring repeat, where IFT140 (IFT140$^{N}$) interacts with the C-terminal TPRs of IFT144 from the adjacent complex (IFT144$^{N-1}$) (Extended Data Fig. 9f,g). This interaction supports the end of IFT144$^{N-1}$-TPR, which acts as the base on which IFT140$^{N-1}$-WD and IFT121$^{N}$-WD sit. This unusual arrangement means that IFT144 and IFT140 are responsible for both lateral interactions and the fundamental structural organization of the neighboring repeat.

IFT122, IFT121 and IFT139 form three pillars at the other end of IFT-A. The IFT122 and IFT121-WD domains are stacked together directly below the membrane. IFT121-TPR runs through this region to form a platform for IFT122-WD binding and slots into the IFT139

superhelix. (Fig. 4a). Finally, IFT122-TPR projects out of the column toward IFT144/140, where it interacts with IFT144-WD (Fig. 4c).

## IFT-A alterations are clustered around interfaces

The Human Gene Mutation Database contains over 100 point mutations that lead to alterations in IFT-A proteins associated with ciliopathy phenotypes[36]. Many of these alterations can be mapped to the outer surfaces of the WD domains in our model (Fig. 4d,e and Supplementary Data 1). Since these regions all face the membrane directly, alterations here could have a deleterious effect on membrane recognition or cargo binding. In IFT144 and IFT140, many of the WD domain alterations correspond to the regions that interact with the unidentified extra density (Extended Data Fig. 9g). This suggests that this extra density could be an IFT-A cargo or cargo adapter.

In the TPR domains, almost all of the alterations are found at the interfaces with other IFT-A proteins (Fig. 4d,e). This includes interactions between IFT144 and IFT140 belonging to neighboring repeats (Fig. 4e). These alterations are therefore likely to result in destabilization of the complex, due to disruption of complex formation or polymerization. IFT139 is an exception because it contains alterations throughout its structure. It forms an external surface, thus alterations are likely to disrupt interactions with cargo or IFT-B (as discussed below) rather than complex formation.

## IFT-A and IFT-B are flexibly tethered

A major remaining question is how IFT-A and IFT-B stably bind to each other, given their periodicity mismatch. In our IFT-A and IFT-B averages, the mismatch meant that one complex was blurred out in the average of the other (Fig. 5a–c). By using masked 3D classification of the region corresponding to IFT-A in our IFT-B averages, we obtained classes where IFT-A is resolved in different registers relative to IFT-B (Extended Data Fig. 10a). In these classes, we see two new densities bridging IFT-A and IFT-B (Fig. 5d,e).

The first bridge is between IFT139 in IFT-A and IFT81/74 in IFT-B1 (Fig. 5d). Each IFT-B1 repeat projects a tubular density corresponding in length and location to the unmodeled fifth coiled-coil segment of IFT81/74. Two IFT81/74 copies bind to one IFT139, although there are transition zones where the periodicity mismatch means that two adjacent repeats compete for the same IFT139 binding site (Fig. 5e). Here, there is a switch in register in the subsequent repeats, made possible by the conformational flexibility between IFT81/74 coiled-coil segments. IFT139 has a negatively charged surface and IFT81/74-CC5 is positively charged, making a favorable ionic interaction possible (Extended Data Fig. 10b,c). The mutations in IFT139 that we find in this region (Fig. 4d) could therefore affect IFT81/74 binding.

The second bridge comes from classes obtained from our IFT-B2 average. We see an extension of the IFT172 density running along the roof of IFT-B2 in alternate repeats (Fig. 5f,g). This density reaches up to the IFT-A complex and docks between the C terminus of IFT144 and the inner face of IFT139. This links IFT-A complexes two repeats away from each other, suggesting that it could be important to help guide IFT-A poylmerization by establishing longer-range lateral interactions. We assign this density to be the C-terminal TPR domain of IFT172, which is also unmodeled in our overall reconstruction. Like IFT81/74-CC5, this domain is linked to the modeled region by a flexible linker, allowing it to interact with IFT-A in different registers. The IFT172 C terminus contains a strongly acidic patch capable of binding to a basic patch on IFT144 (Fig. 5h,i).

Together, we show that anterograde trains overcome the periodicity mismatch between IFT-A and IFT-B using flexible tethers from IFT-B that are in a stoichiometric excess to IFT-A. This suggests that IFT-A is recruited in a search-and-capture mechanism, where nascent IFT-B polymers can sample a large space through these tentacle-like tethers (Fig. 5j,k). This then aids IFT-A polymerization by creating a higher local concentration of IFT-A and promotes long-range lateral

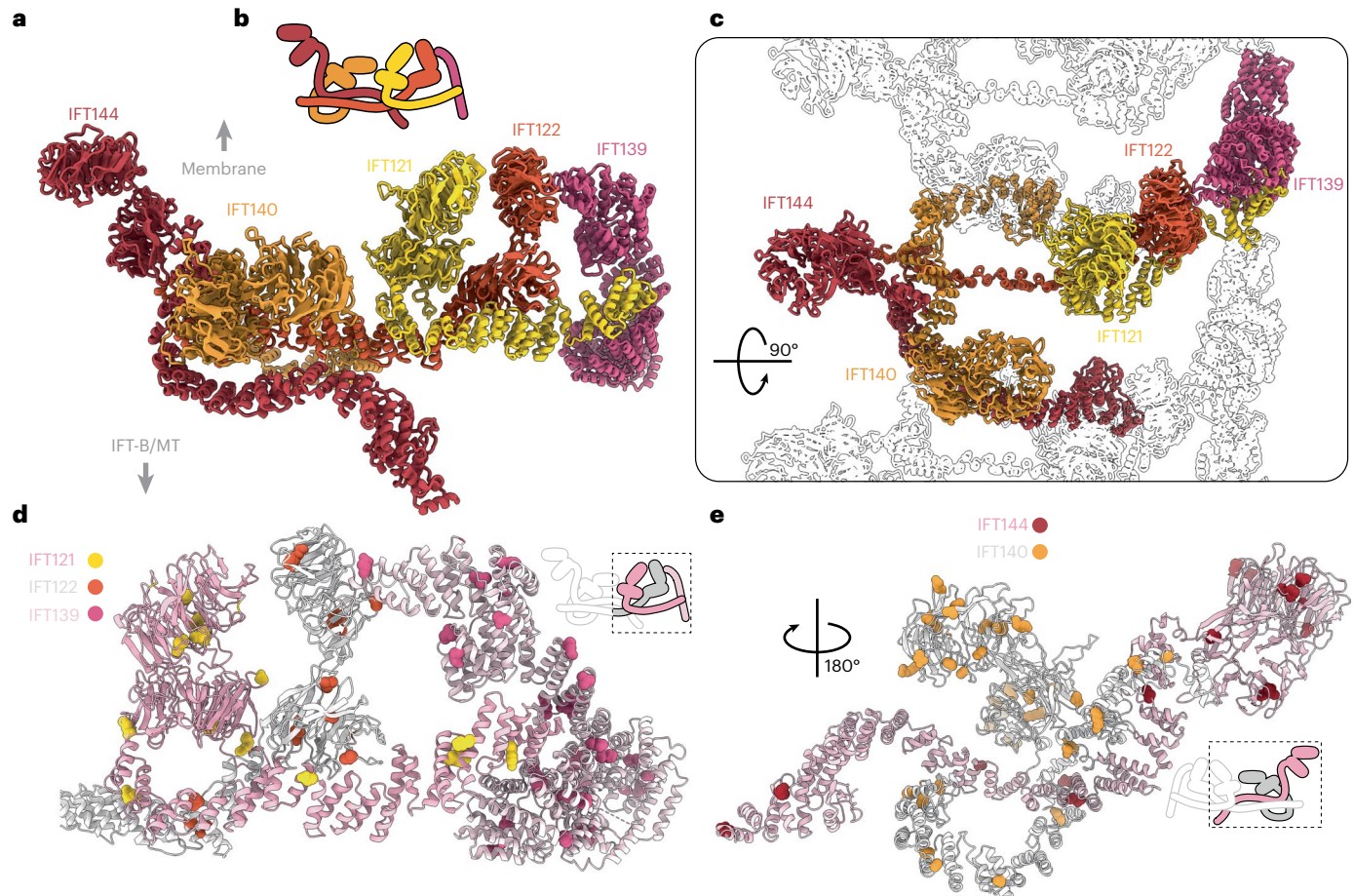

**Fig. 4 | IFT-A presents its four WD domains to the membrane. a,** The IFT-A model viewed in profile, as if looking down the train. **b,** Cartoon representation of IFT-A shown from a side view as in **a**. **c,** Top view of the IFT-A model, with neighboring repeats shown as silhouettes. IFT140 and IFT144 both reach into the neighoring complex. **d,** We mapped alterations in human IFT-A proteins caused by point mutations that are linked to ciliopathies to conserved residues in *C. reinhardtii*. Here, IFT121, IFT122 and IFT139 are shown, with most alterations (shown as sphere representation) mapping to the WD domains or to interfaces between TPR domains. **e,** A second view, showing the alterations caused by point mutations present in IFT144 and IFT140.

interaction into polymers (Fig. 5g). In principle, this could mean that IFT-A could only polymerize with the help of IFT-B, thus preventing IFT-A multimerization away from the basal body. Finally, a flexible interaction allows IFT-A and IFT-B to maintain their connection while withstanding the mechanical stresses present in actively beating cilia.

## Discussion

Overall, we present a complete molecular model of the anterograde IFT train. This was made possible by recent improvements in subtomogram averaging methods and protein structure prediction. The use of Alpha-Fold2 models in combination with intermediate-resolution cryo-ET densities opens many new avenues for previously difficult-to-characterize protein complexes, but is a technique that needs to be treated with caution. Our modeling process was complemented by a wealth of previously published protein–protein interactions that limited the combination of possible protein positions to a single solution (Extended Data Fig. 5). Subsequently released results from a single-particle structure of isolated IFT-A complexes[37] and crosslinking mass spectrometry of purified IFT-B[38] are both consistent with our model.

Our new model finds interactions within anterograde IFT trains that are not described in previous studies. We propose that since the previously mapped interactions are based on purified complexes outside of their native environment, these probably represent isolated, unpolymerized IFT complexes. Differences in interactions between our structure and the previous data could therefore illustrate the

architectural changes that occur during polymerization into anterograde trains.

For example, IFT81/74 was conventionally thought to be recruited to IFT-B1 through interaction with the IFT52/46 heterodimer[11,23]. In our model, IFT81/74 instead docks onto IFT88 and IFT70. In a recent crosslinking mass spectrometry study of purified IFT-B complexes, the presence of the IFT88/70 interaction was detected and it was shown that it is mutually exclusive with the more dominant IFT52/46 interaction[38]. This suggests that during polymerization into anterograde trains a conformational change occurs in IFT-B1 that stabilizes the second IFT88/70 binding site.

In IFT-B2, IFT172 and IFT80 were previously shown to only interact in the TPR regions[10,39]; however, our model shows that the WD domains also form part of the interface. These interactions occur across the interface between adjacent repeats, meaning that they are unlikely to be detected after purification for coimmunoprecipitation assays. This is consistent with data showing that purified IFT-A and IFT-B complexes do not oligomerize, even at high concentrations[11,37]. This leads to a conundrum of how the IFT-B polymer is assembled when the interactions forming lateral repeats are too weak to be detected biochemically. One possible answer could be that an exogenous factor is required to nucleate or assist polymerization. Interestingly, in subtomogram averages of anterograde trains assembling at the basal body, an unknown extra density is observed beneath IFT-B1 that is absent in the mature train[3]. This unknown component could therefore be responsible for

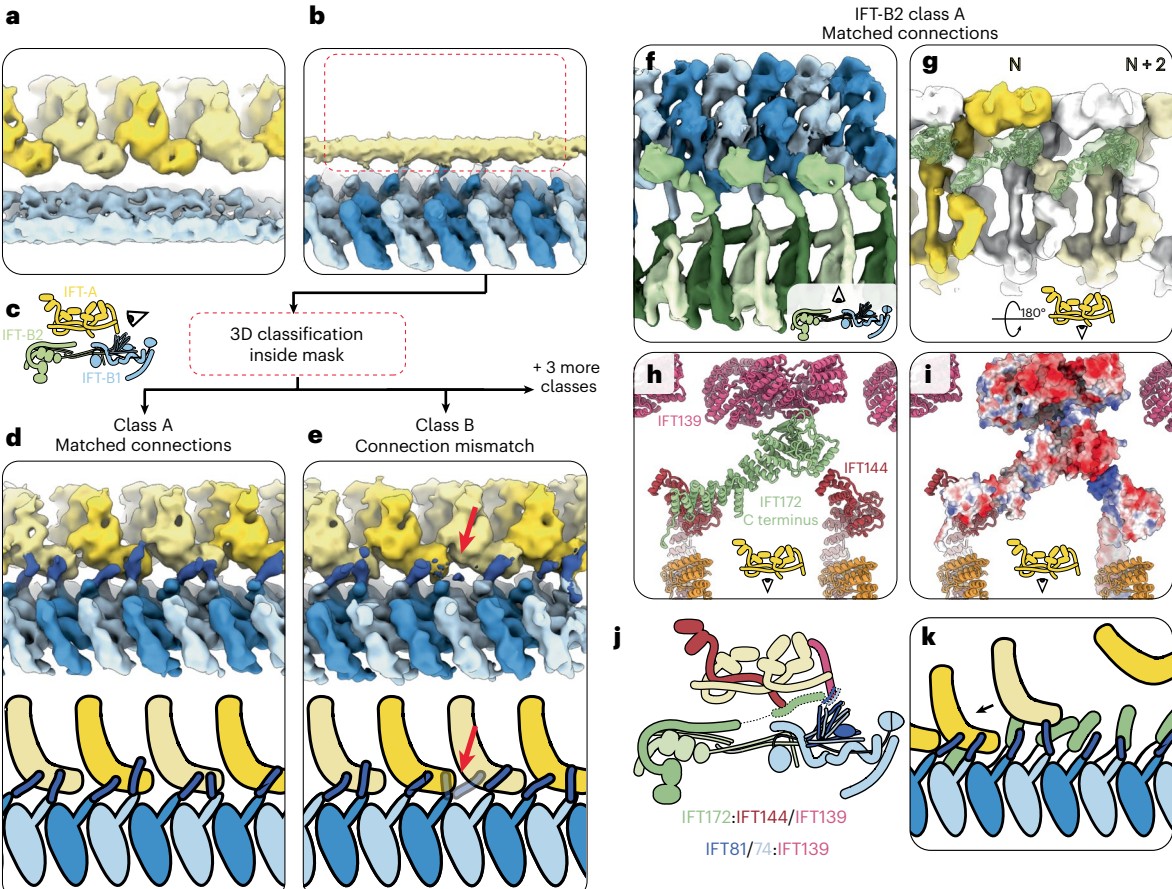

**Fig. 5 | IFT-A and IFT-B are connected at two points. a**, The 21 Å IFT-A average covering three repeats, unmasked to show that IFT-B (light blue) is averaged out with respect to IFT-A (alternating yellow) due to peridocity mismatch. **b**, The IFT-B1 average filtered to 12 Å and unmasked, to show that IFT-A (yellow) is averaged out with respect to IFT-B1 (alternating blue) due to periodicity mismatch. The red box indicates the location of the mask used for subclassification to generate the classes in **d** and **e**. **c**, Cartoon depicting the view in **a**, **b**, **d** and **e**. **d**, After classification of the IFT-A region in the IFT-B1 average, we find classes where IFT-A (alternating yellow) and IFT-B (alternating blue) are in sync. We see a new density (dark blue) linking IFT-B to IFT-A, which we designate as CC5 of IFT81/74. Bottom, cartoon representation of the density. **e**, A second class shows how the IFT81/74 connections (dark blue) adapt to the periodicity mismatch between IFT-A (alternating yellow) and IFT-B (alternating blue), by switching register with respect to IFT-A at the red arrow. Bottom, cartoon representation of the density. **f**, A top view of class A from classification

of the IFT-A region in the IFT-B2 average. Inset, cartoon view. IFT-B1 (alternating light/dark blue) and IFT-B2 (alternating light/dark green) are joined by a new, unmodeled density corresponding to the C terminus of IFT172 (lime green). **g**, The same class as **f**, rotated 180° to view the same IFT172 density (lime green and transparent, with the AlphaFold2 model docked) interacting with IFT-A. The IFT-A complex is colored to highlight that the connecting density connects nonadjacent neighbors. Inset, cartoon view. **h**, The same view as in **g**, showing the AlphaFold2 IFT172 C terminus model (lime green) docked into the density along with our IFT-A model. IFT172 bridges the gap between IFT144 and IFT139. **i**, The same view as in **h**, with IFT172, IFT144 and IFT139 shown with surface charge depiction. The negatively charged IFT172 C terminus can make favorable ionic interactions with the positively charged IFT144 C terminus. **j**, Cartoon representation of the overall anterograde train structure, showing the two points of connection (dotted outlines). **k**, Cartoon representation depicting the proposed role of the flexible tethers in recruiting IFT-A complexes to nascent IFT trains.

starting the process of fixing mobile domains into a single conformation during polymerization.

Finally, the connection between IFT-A and IFT-B had recently been shown to be mediated by an interaction between the C terminus of IFT88 in IFT-B1 and the C terminus of IFT144 in IFT-A[37,40]. These two elements are close enough in our model to interact, although we do not have the resolution in this region to detect the contact. However, since the IFT88 C terminus is long and disordered, it lacks the structural rigidity to tether IFT-A to IFT-B in the tight interaction seen in anterograde trains. The IFT88–IFT144 interaction could therefore represent the first contact in a multistep recruitment process, in which a loose initial attachment is followed by the tighter tethering we observe to achieve the mature anterograde structure.

A key outstanding question is how the structure we show here remodels into the conformationally distinct retrograde train. We recently showed that anterograde-to-retrograde train conversion

in *Chlamydomonas* can be induced by mechanical blockage of IFT at arbitrary positions along the length of the cilium[41]. This indicates that anterograde-to-retrograde remodeling does not require specialized machineries of the ciliary tip. This supports a model in which conversion occurs through conformational changes prebuilt into the anterograde train. This could be through the compressed or spring-like coiled coils such as IFT81/74 or IFT57/38. Alternatively, TPR and other α-solenoid domain proteins have previously been shown to behave as molecular springs[42–44]. Many of the TPR domains in our structure underwent curved-to-straight conformation changes to fit the relaxed AlphaFold2 predictions into our density (Extended Data Fig. 4b), indicating that they could be a source of molecular strain. This strain could then be released at the tip, potentially triggered by the loss of tethering to the microtubule, resulting in a relaxation into the retrograde conformation. However, to fully understand how train conversion occurs, more structural information of the retrograde train is required.

## Online content

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

## Methods

### Cell culture

*C. reinhardtii* wild-type (CC625) cells and CC625 cells with glycocalyx proteins FMG1A and FMG1B deleted by CRISPR (produced for and described in a manuscript by Nievergelt and Pigino, in preparation) were cultured in aerated Tris-acetate-phosphate (TAP) media at 24 °C with a 12 h night/12 h dark cycle for at least 2 d before use.

### Grid preparation

Quantifoil R3.5/1 Au200 grids were plasma cleaned for 10 s with an 80:20 oxygen:hydrogen mix (Solarus II Model 955; Gatan). Then, 4 µl cells were added to the grid, followed by 1 µl 10 nm colloidal gold fiducial solution (in phosphate-buffered saline; BBI Solutions). Following 30 s incubation at 22 °C and 95% humidity, the grid was back-blotted and immediately plunge frozen in liquid ethane at −182 °C (Leica Automatic Plunge Freezer EM GP2).

### Cryo-ET data acquisition

Cryo-ET data were acquired on a Thermo Scientific Titan Krios G4 transmission electron microscope operated at 300 kV using SerialEM[45]. Raw video frames were recorded on a Thermo Scientific Falcon 4 direct electron detector using the post-column Thermo Scientific Selectris X energy filter. Videos were acquired in Electron Event Representation format[46] with a pixel size of 3.03 Å per pixel, an exposure of 3 s and a dose rate of $2.6e^-$ $Å^{-2}$ $s^{-1}$. Tilt series were collected in 3° increments using a dose-symmetric scheme with two tilts per reversal up to 30°, and then bidirectionally to 60°. For a full tilt series, this resulted in an accumulated dose of $104e^-$ $Å^{-2}$. Tilt series were acquired between −2.5 and −4.5 µm defocus.

### Tomogram reconstruction

Tilt series reconstruction was performed using a developmental update of the TOMOMAN pipeline[47], which organizes tomographic data while feeding it into different preprocessing programs. Motion correction was performed using the MotionCor2 implementation in Relion 3.1 (ref. 48), with Electron Event Representation data split into 40 fractions. Bad tilts were then removed after manual inspection, followed by dose weighting (Imod[49]) and contrast transfer function (CTF) estimation (CTFFIND4; ref. 50). Manual fiducial alignment and CTF-corrected tomogram reconstruction at bin4 were then performed in Etomo[49]. The bin4 tomograms were then deconvolved for visualization with the tom_deconv filter[51].

### Particle picking

Anterograde IFT trains were identified in deconvolved bin4 tomograms according to features identified previously[9]. Picking was performed using the 3DMOD slicer[49], with IFT-B and IFT-A picked separately. For each IFT-B and IFT-A filament, an open contour model was picked along the length. Points were picked along this contour at 4 and 2 nm distances for IFT-A and IFT-B, respectively (representing an oversampling of ~3× in each case) using TOM Toolbox scripts (https://www.biochem.mpg.de/6348566/tom_e).

### Subtomogram averaging

We used STOPGAP[52] to find initial orientations before transferring data to Relion for high-resolution refinements. However, we found that because IFT-B looks similar with 180° rotation around the long axis (the phi angle in STOPGAP) the initial angles were split roughly 50/50 with the right and wrong phi angle. We therefore analysed each train individually and determined a rough phi angle manually. In STOPGAP, we extracted particles from the unfiltered bin4 tomograms (70 and 50 pixel box sizes for IFT-B and IFT-A, respectively) and performed alignments using a cone search with a 32° phi search in 8° increments.

The particles and orientations from STOPGAP were converted to Relion star format and subtomograms and 3D CTF particles were extracted in Warp[53].

For IFT-B, six different collection sessions were incrementally added to the average (Extended Data Fig. 2). Each group was refined separately in STOPGAP, with the STOPGAP average of the first group used as the initial reference for 3D refinement in Relion 3.1 (ref. 48). Initial refinements used a solvent mask consisting of the entire IFT-B complex for four repeats. We performed a local 3D refinement with 3.7° initial angular sampling per step and 4 and 1 pixel initial translational search and step sizes. The resulting refinement was used as the input for a round of image warp grid refinement in M[54]. The refined subtomograms were re-extracted and the 3D refinement was repeated, resulting in a greatly improved average. This refinement was then used as the input for 3D classification into two classes, using the same solvent mask and keeping the alignments fixed. The particles from the good class were then used for separate masked refinements of IFT-B1 and IFT-B2, which proceeded independently but with the same input particles. For IFT-B1, we found that reducing the length of the mask to two repeats resulted in the best averages, but IFT-B2 was best at four repeats. Both subcomplexes reached Nyquist resolution, so IFT-B1 was re-extracted eventually to bin 1 (3.03 Å per pixel) and IFT-B2 was re-extracted to bin 1.5 (4.04 Å per pixel). We obtained the highest-resolution reconstructions after performing image warp and CTF refinement on the IFT-B1 reconstruction in M. We used the resulting parameters to re-extract both IFT-B1 and IFT-B2 particles for a final round of 3D refinement (1.7° initial angular sampling; 3/1 pixel initial translational search/step). The resolution was determined with the 0.143 threshold (Extended Data Fig. 3a,b). Masked refinement of the ends of IFT-B1 and IFT-B2 resolved these regions more clearly, although still at lower overall resolution compared with the core masks (Extended Data Fig. 2c). To obtain an average of dynein, we created a solvent mask based on our previous low-resolution IFT-B/dynein average and rescaled it to 4.04 Å per pixel (Extended Data Fig. 2d). We performed 3D classification on our IFT-B2 average into six classes without refinement (Extended Data Fig. 2a), finding three classes with dynein in three registers. We selected one class and performed local refinement.

For IFT-A, the six collection session groups were combined directly after STOPGAP into a local refinement in Relion using a mask with three repeats (Extended Data Fig. 4). We did not perform image warp refinement in M for IFT-A as it resulted in a worse average compared with when the refinements from IFT-B1 were used. However, we found that after the first refinement in Relion, we saw a strong improvement by applying the median Phi angle for each train to every particle in the same train (coordinate smoothing). This pulls particles that have strayed back to the consensus angle for the train. The smoothed coordinates were then locally refined in Relion again and this refinement was used for masked 3D classification without alignments. The good class re-extracted at bin2 (6.06 Å per pixel) and locally refined with a selection of masks (one repeat, three repeats, left side and right side; Extended Data Fig. 4b–e) to generate maps that best show individual features within the complex and also connections between adjacent complexes.

### Model building

A number of crystal structures were available for IFT-B components, but we used AlphaFold2 structural predictions for all of the components because the crystal structures were either from different species or only contained fragments of the protein. Structure predictions were run as monomers or multimers using a local install of AlphaFold version 2.1.1 (ref. 55). AlphaFold2 predictions exhibited no major differences compared with the solved crystal structures. All IFT-A proteins were folded as monomers. For IFT-B, IFT172 and IFT56 were the only proteins folded as monomers. In IFT-B1, the complexes folded as multimers were IFT88/52/70, IFT70/52/46 (ref. 11) and IFT81/74 (ref. 13). For IFT70, the best fit of the density was achieved by splitting the model in two, with the IFT88/52/70 prediction contributing the C terminus and the IFT70/52/46 prediction contributing the C terminus. IFT52 was split at the same place as IFT70. In IFT-B2, we folded IFT80/57/38 and IFT54/20 as multimers[10,15].

Once we had these starting models, the position of most of the IFT-B proteins in the density was straightforward. IFT172, IFT88/70/52, IFT81/74 and IFT80 all contained strong structural motifs that let us position the original AlphaFold2 models unambiguously. This left the two coiled-coil densities in IFT-B2 to fill. Based on the known interaction between IFT80 and IFT38-CH, we pinpointed the IFT38-CH domain to the density bound to the face of IFT80-WD1. From here, the length of the three IFT57/38 coiled-coil segments exactly matched the coiled-coil density that reaches across from IFT-B2 to IFT-B1. Finally, the length of IFT54/20 matched the coiled-coil density running down the side of IFT80, consistent with the unstructured IFT54 N terminus interacting with cytoplasmic dynein-2.

For IFT-A, the four proteins with WD domains each contain unique conformations regarding the angle between the tandem WD domains and between the second WD domain and the start of the TPR. This allowed us to place each of the four WD domains into the density unambiguously. We recognized that the proteins could not adopt reasonable conformations to fit into one repeat as defined in our previous cryo-ET structure. However, we could identify continuous density between adjacent repeats in the average of three consecutive IFT-A repeats. The IFT139 TPR superhelix was obviously identifiable at the edge of the complex, but was split into two rigid bodies at a loop in the middle of the protein to best fit the density.

Once we had positioned the models in the density, we manually edited them to best fit the density. In IFT-B1, in regions where individual α-helices were resolved (IFT88, IFT70, IFT81/74 and IFT57/38), this involved conventional secondary structural real-space refinement in Coot[56]. In IFT-B2, the IFT54/20 coiled coil needed to be curved slightly to fit into the density. The C-terminal TPR domains of IFT172 curved out of the density. To counter this, we split the region into rigid bodies defined by loops where the AlphaFold2 prediction had lower confidence. We then fit the rigid bodies up to the point where the density became too weak, leaving roughly one-third of IFT172 unmodeled (Extended Data Fig. 4b). We used the same approach for the TPR domains in IFT-A. For IFT140, IFT122 and IFT121, we did not model the flexible TPR regions at the C termini. This is because they were predicted to be only loosely tethered to the remaining TPR regions, but in each case there is empty density left in the average for them to occupy.

Once we had manually assembled the models into the density, we used NAMDinator[57], an automated molecular dynamics flexible fitting pipeline, to refine to models into our density. We used default parameters and started with the individual assemblies described above. Different models were then combined to form the IFT-B1/2 and IFT-A complexes and refined, and then combined again to create lateral repeats to ensure lateral did not clash. Map and model visualization were performed in ChimeraX[58]. Human point mutations were obtained from the Human Gene Mutation Database[36].

### Reporting summary

Further information on research design is available in the Nature Portfolio Reporting Summary linked to this article.

### Data availability

The following maps have been deposited to the Electron Microscopy Data Bank: the IFT-B consensus of focused refinements (EMD-15977), the IFT-B1 focused refinement (EMD-15978, with the IFT-B1 peripheral focused refinement as an associated map), the IFT-B2 focused refinement (EMD-15979, with the IFT-B2 peripheral focused refinement as an associated map), the IFT-B low-resolution overall map to validate consensus (EMD-16014) and the IFT-A three-repeat map (EMD-15980, with one-repeat and masked refinements as associated maps in this deposition). The IFT-B and IFT-A atomic models have been deposited to the Protein Data Bank with the codes 8BD7 and 8BDA, respectively.

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

## Acknowledgements

We thank P. Swuec and S. Sorrentino (from the Human Technopole electron microscopy facility), C. Fernandez and P. Margara for IT and high-performance computing support, P. Erdmann and S. Khavnekar for providing the TOMOMAN and STOPGAP implementations, D. Diener, A. Vanninni and F. Coscia for comments on the manuscript and A. Nievergelt (Max Planck Institute of Molecular Cell Biology and Genetics) for CRISPR-modified cell lines. We acknowledge funding from Human Technopole and the European Research Council under the European Union's Horizon 2020 Research and Innovation Programme (grant agreement number 819826) to G.P. and EMBO ALTF 1141-2021 to H.E.F.

## Author contributions

S.E.L. prepared the samples, acquired the cryo-ET data, performed image processing, refined, analysed and interpreted the data and wrote the manuscript. H.E.F. performed AlphaFold2 structural predictions. G.P. designed the experiments, interpreted the data and wrote the manuscript.

## Competing interests

The authors declare no competing interests.

## Additional information

**Extended data** is available for this paper at https://doi.org/10.1038/s41594-022-00905-5.

**Correspondence and requests for materials** should be addressed to Gaia Pigino.

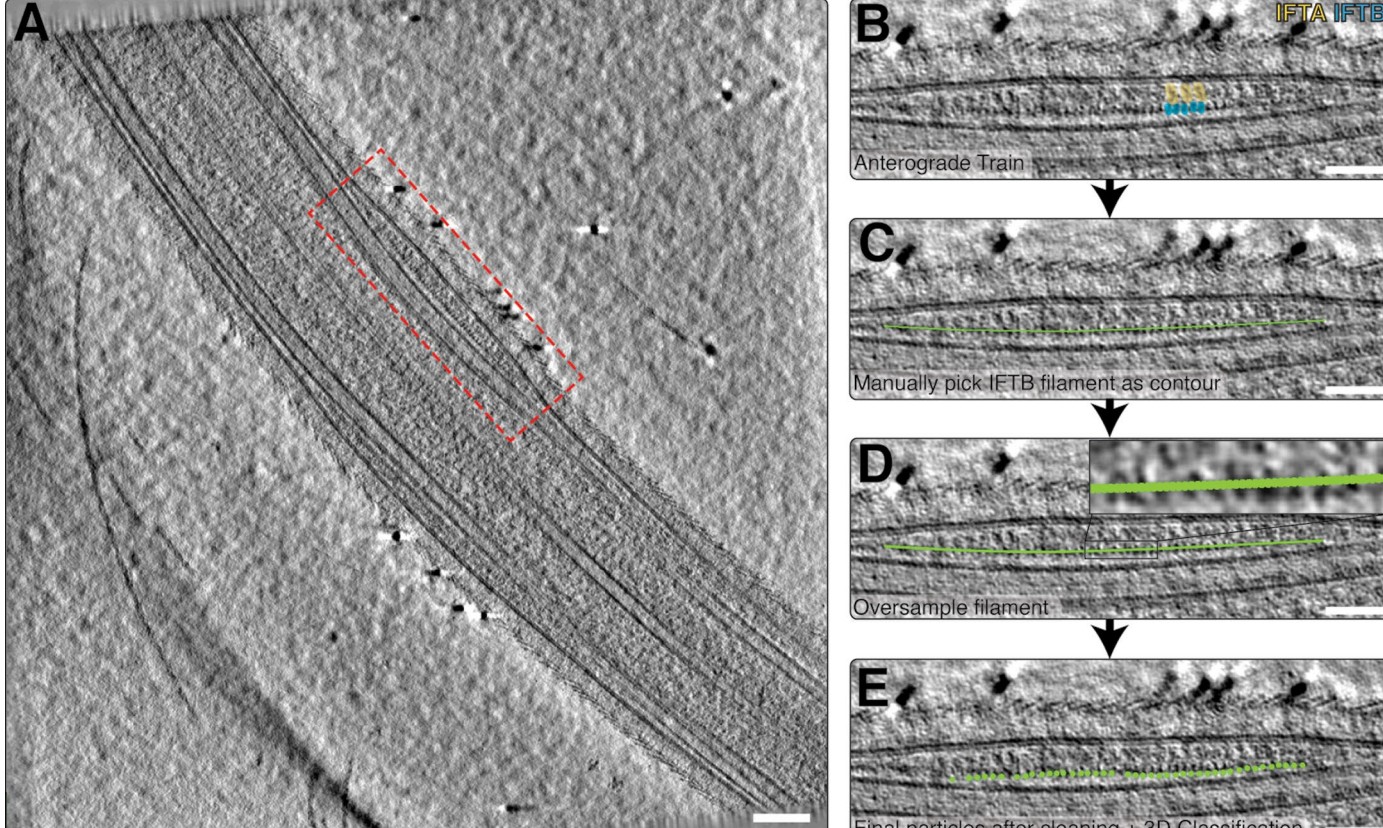

**Extended Data Fig. 1 | Identification of anterograde IFT trains in cryo-electron tomograms. a**, A slice through a representative tomogram from our dataset of a 600 tomograms of C. reinhardtii cilium, showing a bulge in the membrane in the middle corresponding to an anterograde IFT train (red box). Scale bar = 100 nm. **b**, Close up view of the train in A, with IFT-A (yellow) and IFT-B (blue) repeats annotated. Scale bar = 50 nm. **c**, After identification, we manually picked trains in IMOD as a contour running through the center of the complex. IFT-B picking is shown here, and IFT-A, visible above the IFT-B contour, was picked in a separate model. Scale bar= 50 nm. **d**, The contour was converted into subtomogram coordinates with oversampling to ensure no particles were missed. Scale bar= 50 nm. **e**, Here, the final refined coordinates are shown on the train. The particles have undergone proximity cleaning compared to the oversampling in D, as well as 3D classification to remove bad particles. Scale bar = 50 nm.

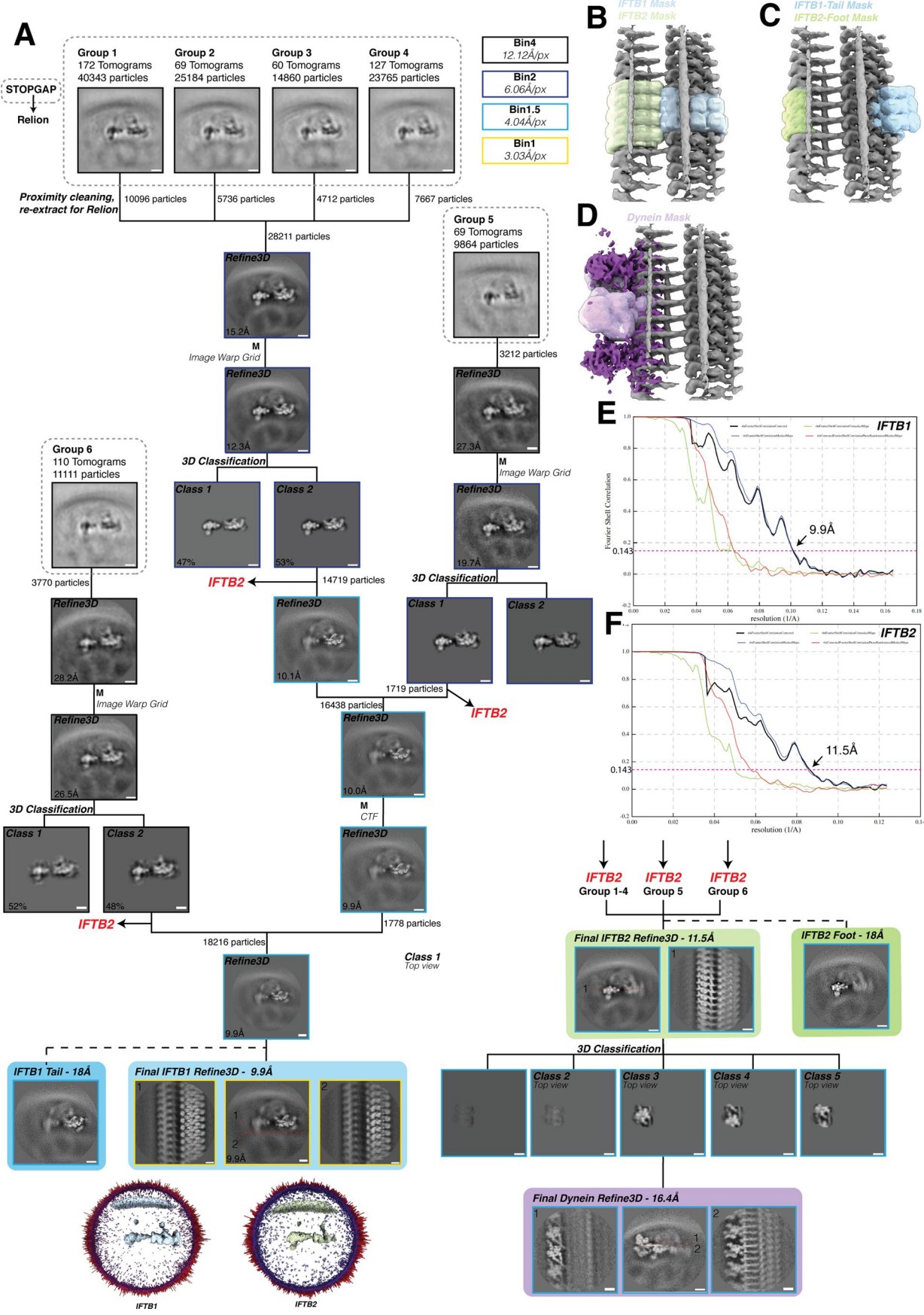

**Extended Data Fig. 2 | See next page for caption.**

**Extended Data Fig. 2 | Processing diagram for IFT-B subtomogram averaging.**
**a**, Workflow depicting the steps involved in averaging the IFT-B1 and IFT-B2 complexes. Processing started in STOPGAP (areas in dotted black line) before proceeding to Relion. The level of binning at each stage is indicated by the outline of the box (colour code top right). All scale bars=10 nm. **b**, The solvent masks used to refine IFT-B1 (blue) and IFT-B2 (green) separately from each other. **c**, The solvent masks used to refine the extremities of the IFT-B1 and IFT-B2 complexes, which are poorly resolved when using the masks in B. **d**,The solvent mask used to classify and refine dynein from IFT-B2. **e**, Fourier Shell Coefficient (FSC) curve of the IFT-B1 average, as a measure of map resolution. **f**, FSC curve of the IFT-B2 average.

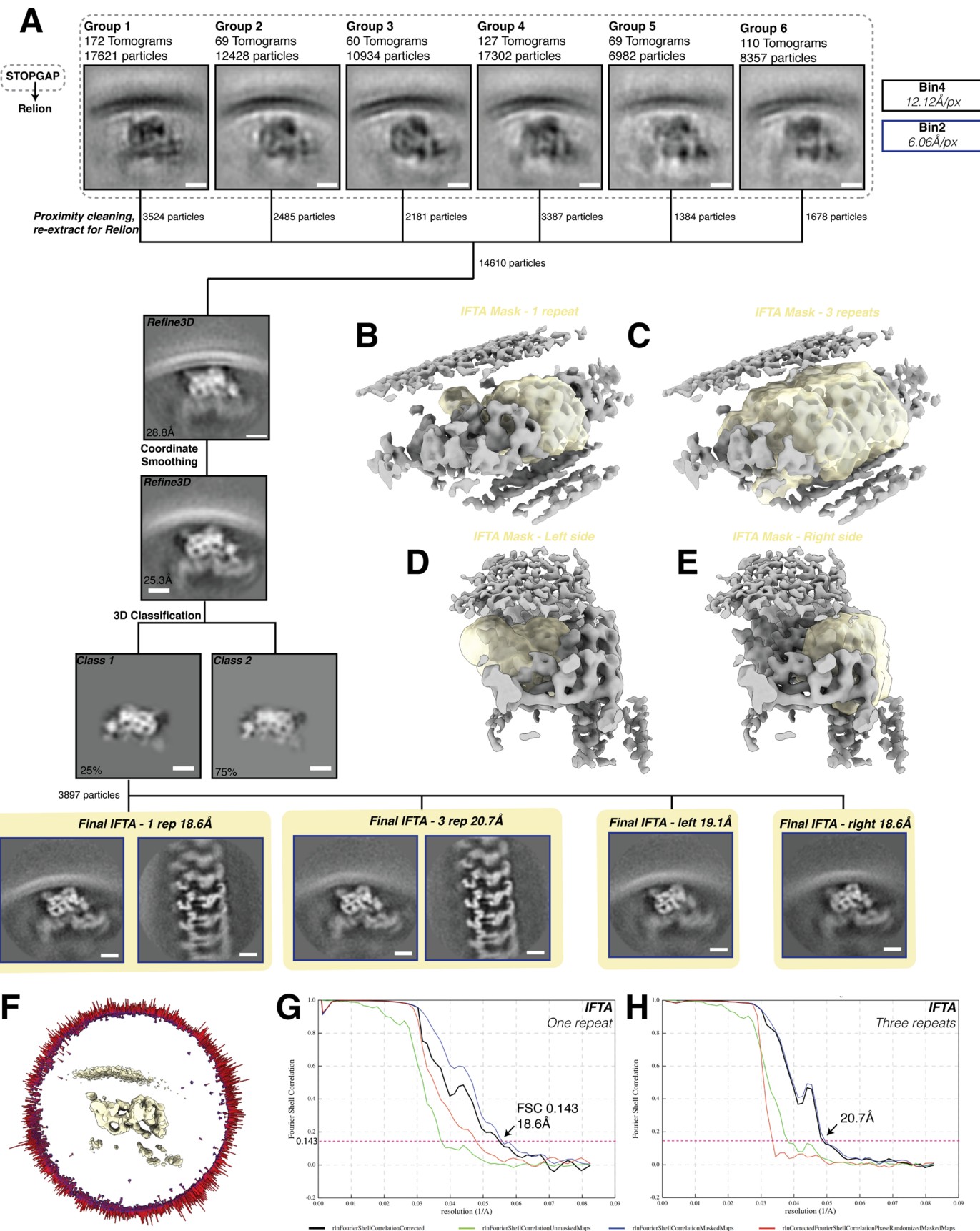

**Extended Data Fig. 3 | See next page for caption.**

**Extended Data Fig. 3 | Processing diagram for IFT-A subtomogram averaging.**
**a**, Workflow depicting the steps involved in averaging the IFT-A complex.
Processing started in STOPGAP (areas in dotted black line) before proceeding
to Relion. The level of binning at each stage is indicated by the outline of the box
(colour code top right). All scale bars=10 nm. **b**, The solvent mask used to refine
IFT-A, containing one repeat. **c**, The solvent mask used to refine IFT-A, containing
three repeats. **d**, The solvent mask used to refine IFT-A, consisting of the left side
of one repeat of the complex. **e**, The solvent mask used to refine IFT-A, consisting
of the right side of one repeat of the complex. **f**, Angular distribution of particles
contributing to the IFT-A average (one repeat). **g**, FSC curve of the IFT-A average,
refined using a mask containing one repeat. **h**, FSC curve of the IFT-A average,
refined using a mask containing three repeats.

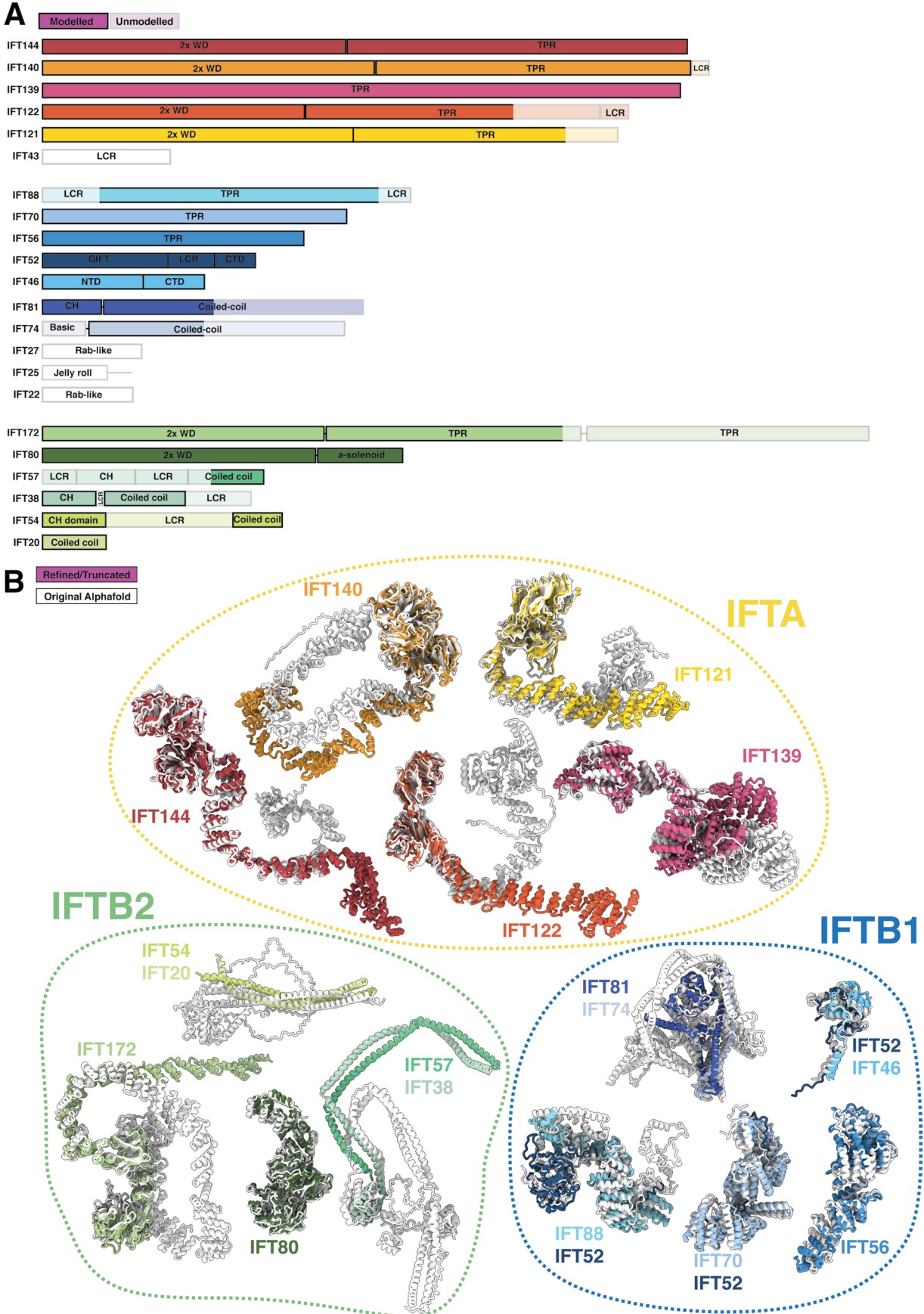

**Extended Data Fig. 4 | Alphafold2 models of IFT components. a,** Domain organization of all IFT constituents. Lighter shading indicates regions that were flexible and unmodelled in our structure. WD = WD40 repeat domain, TPR = Tetratricopeptide repeat domain, CH = Calponin homology domain,

LCR = low-complexity (disordered) region. **b,** The original, unmodified alphafold structures (white) overlaid with the final refined models in our new structure (colours). Refined models have had flexible regions deleted.

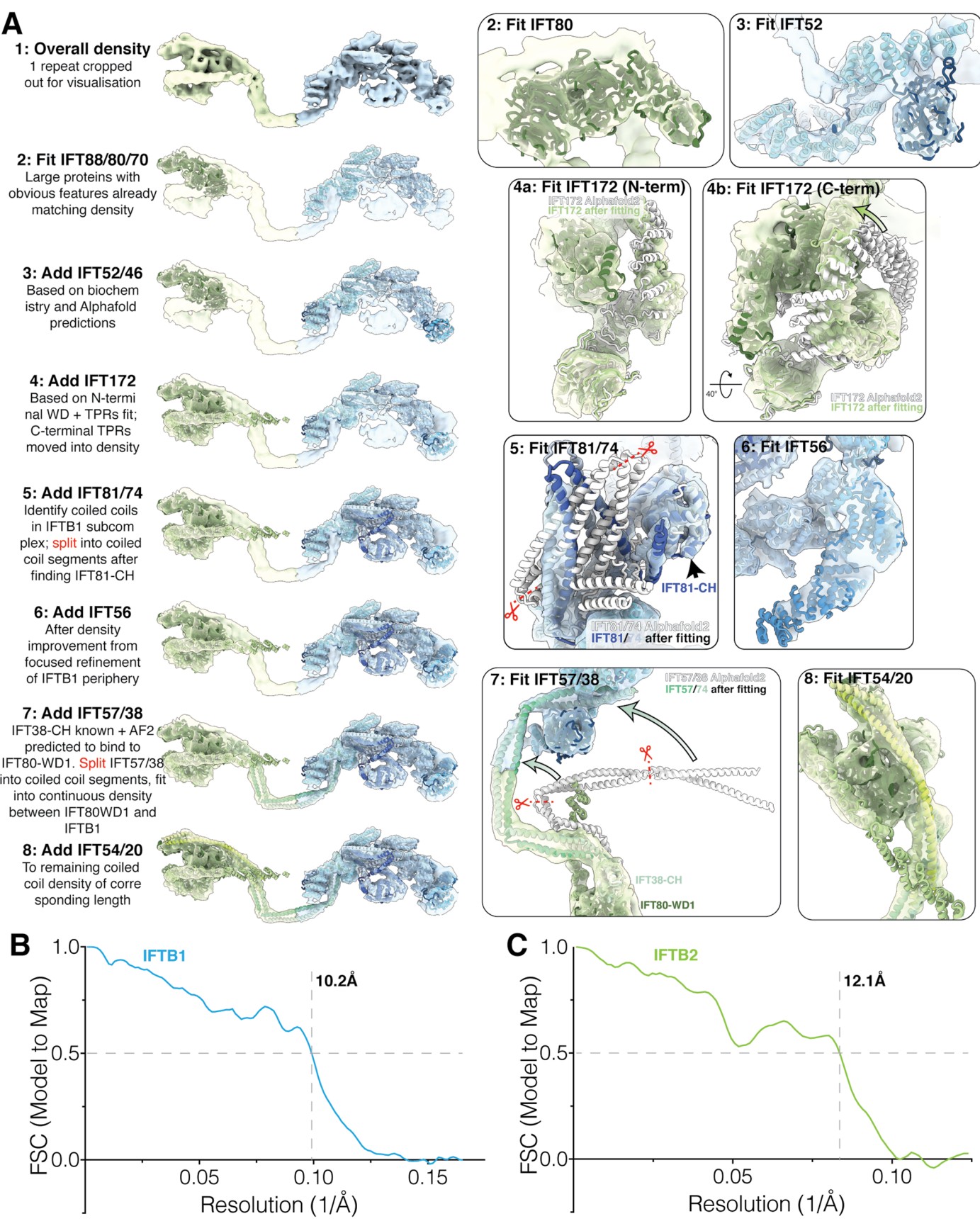

**A**

**1: Overall density**
1 repeat cropped out for visualisation

**2: Fit IFT88/80/70**
Large proteins with obvious features already matching density

**3: Add IFT52/46**
Based on biochem istry and Alphafold predictions

**4: Add IFT172**
Based on N-termi nal WD + TPRs fit; C-terminal TPRs moved into density

**5: Add IFT81/74**
Identify coiled coils in IFTB1 subcom plex; split into coiled coil segments after finding IFT81-CH

**6: Add IFT56**
After density improvement from focused refinement of IFTB1 periphery

**7: Add IFT57/38**
IFT38-CH known + AF2 predicted to bind to IFT80-WD1. Split IFT57/38 into coiled coil segments, fit into continuous density between IFT80WD1 and IFTB1

**8: Add IFT54/20**
To remaining coiled coil density of corre sponding length

**2: Fit IFT80**

**3: Fit IFT52**

**4a: Fit IFT172 (N-term)**
IFT172 Alphafold2
IFT172 after fitting

**4b: Fit IFT172 (C-term)**
IFT172 Alphafold2
IFT172 after fitting
40°

**5: Fit IFT81/74**
IFT81-CH
IFT81/74 Alphafold2
IFT81/74 after fitting

**6: Fit IFT56**

**7: Fit IFT57/38**
IFT57/38 Alphafold2
IFT57/74 after fitting
IFT38-CH
IFT80-WD1

**8: Fit IFT54/20**

**B**  IFTB1
FSC (Model to Map)
1.0
0.5
0.0
10.2Å
0.05   0.10   0.15
Resolution (1/Å)

**C**  IFTB2
FSC (Model to Map)
1.0
0.5
0.0
12.1Å
0.05   0.10
Resolution (1/Å)

**Extended Data Fig. 5 | See next page for caption.**

**Extended Data Fig. 5 | Building a model of IFT-B using Alphafold2 predictions. a**, A step-by-step summary of the placement of each protein in IFT-B during molecular modelling, with accompanying illustrations shown in boxes on the right. *1*: A single repeating unit of IFT-B cropped out of the overall composite map for visualization. IFT-B1 in blue and IFT-B2 in green. *2*: We start by docking in unmodified Alphafold2 models of IFT88, IFT80 and IFT70, which have strong features and required few modifications to the Alphafold2 model *3:* IFT52 was separately folded as a multimer with IFT88, IFT70 and IFT46 based on previous biochemical and structural data. The segments were joined back together and fit into the matching density. *4*: IFT172 was initially identified through the strong fit between the density and the N-terminal WD-domains and TPRs in the Alphafold2 prediction (inset 4a), but the C-terminal TPR domains started to bend out of the density (inset 4b). We therefore moved the TPR domains into the continuous density emanating from the WD domains (arrow, inset 4b). *5*: We concluded that the segmented coiled coil density on the top of IFTB1 was IFT81/74 based on

previous studies. To fit the segments, we split them at the interconnecting loops (red scissor, as well as one more not shown in this view), fit them independently and then reconnected them. *6:* IFT56 was docked in unchanged to the focused refinement of the periphery of IFTB1. *7*: To place IFT57/38, we used the prior knowledge that IFT38-CH forms a high-affinity interaction with IFT80-WD1, and that IFT57/38 forms the link between IFT-B1 and IFT-B2. The linking density between the two lobes is a segmented coiled coil, matching the Alphafold2 prediction of IFT57/38. We therefore placed IFT38-CH in the small globular density bound to the face of IFT80-WD1, and split and docked the coiled coil segments into the bridging density. *8:* IFT54/20 was the remaining Alphafold2 model to fit, and was docked into the coiled coil density of corresponding length in IFT-B2, the only region of the map left unmodelled. **b,c**, Model-to-map FSC curves for the IFT-B1 model into the IFT-B1 density and IFT-B2 model to IFT-B2 density respectively.

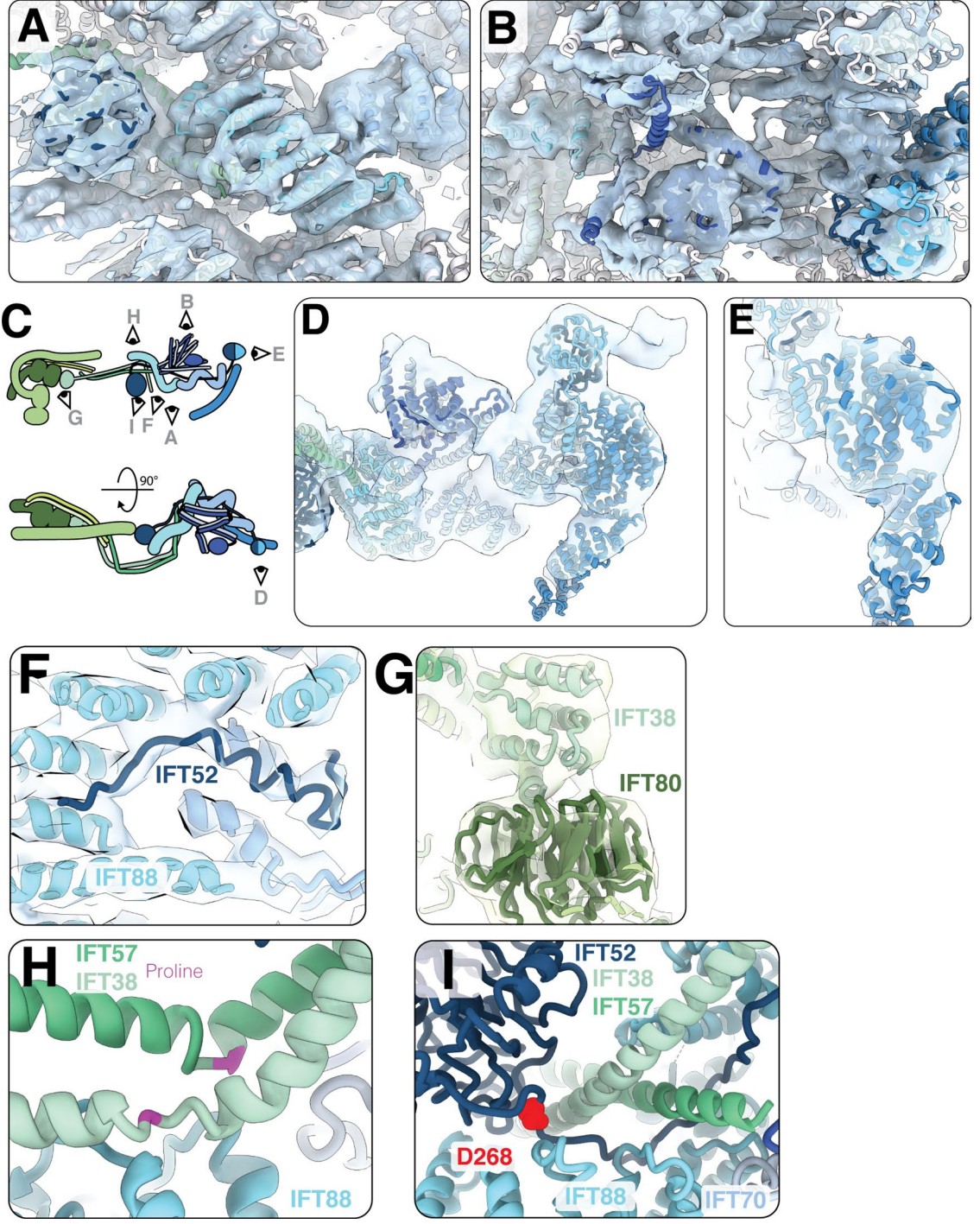

**Extended Data Fig. 6 | Building a model of IFT-B1. a**, A view of the IFT-B1 model docked into its density from the bottom (see E). **b**, A view of the IFT-B1 model docked into its density from the top (see B). **c**, Cartoon representation of IFT-B showing the views in A-D. **d**, A side view of the 'tail' of IFT-B1 docked into the masked tail refinement (Extended data 2A) map lowpass filtered to 18 Å. The region containing IFT56 was more flexible in the high-resolution average shown in A/B, but is more clearly resolved here. **e**, A close up view of IFT56 in the masked tail refinement map, showing that the twist in the TPR helix is visible. **f**, Density for the central unstructured domain of IFT52 (dark blue) is visible in the central pore of IFT88 (cyan), showing that the Alphafold2 prediction agrees with our experimental data. **g**, The N-terminal CH domain of IFT37 (light green) docks to the exterior face of the first WD domain of IFT80 (dark green) in IFT-B2. **h**, A proline residue (magenta) creates a kink in each of the IFT57/38 (dark/light green) helices near the contact to the first IFT88. **i**, The position of D268 in IFT52 highlighted in red, at the interface between IFT-B1 and IFT-B2. D268 in C. reinhardtii corresponds to the D259H mutation in humans[22].

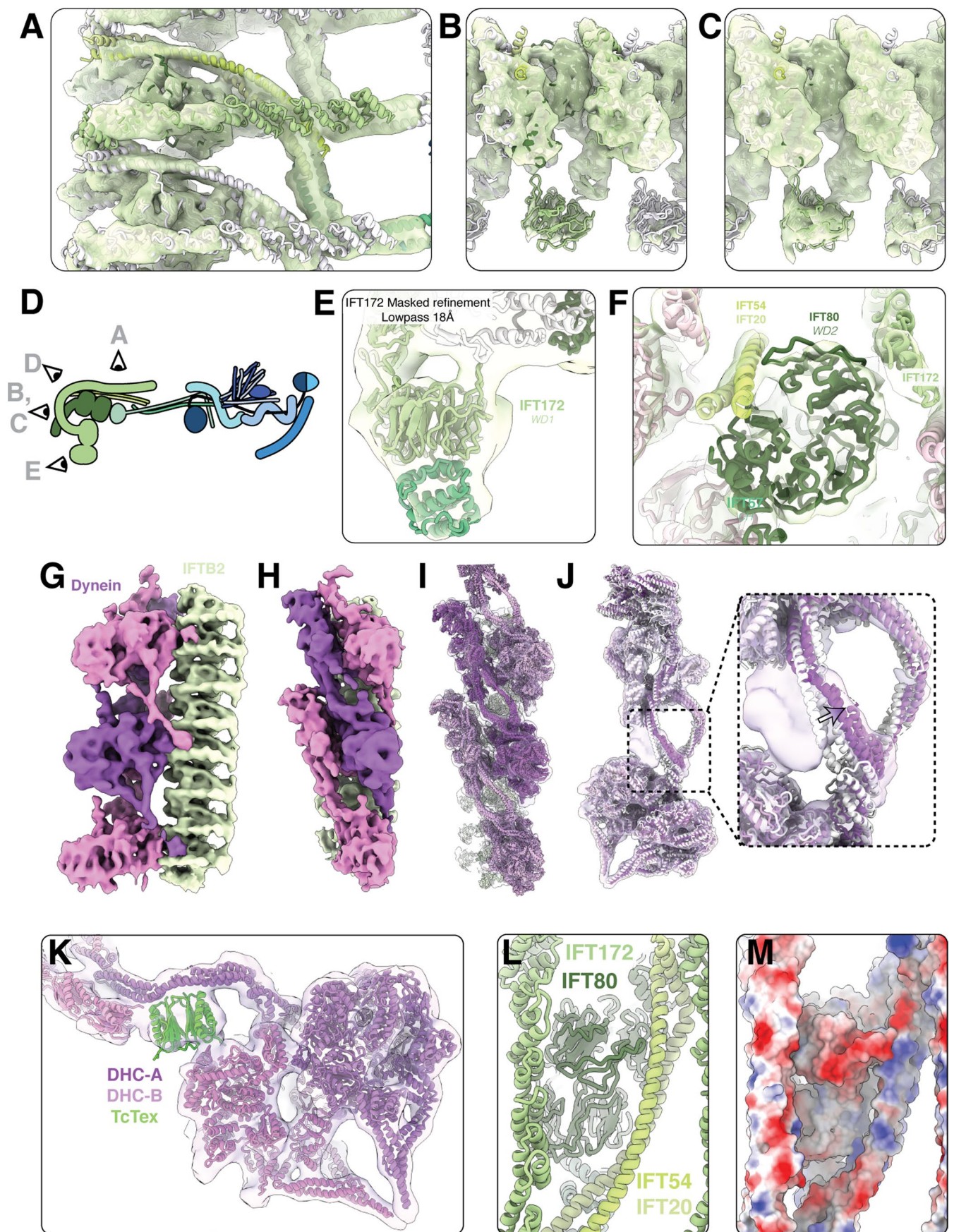

**Extended Data Fig. 7 | See next page for caption.**

**Extended Data Fig. 7 | Building a model of the IFT-B2 complex and its interaction partner dynein-2. a**, A top view of the IFT-B2 subtomogram average density with the IFT-B2 model docked in. **b**, A view of the end of the IFT-B2 subtomogram average density with the IFT-B2 model docked in. **c**, The same view as B, but at a lower threshold to demonstrate that IFT172-WD1 is represented in the density but at lower resolution than the rest of the complex due to flexibility. **d**, Cartoon depicting the views of IFT-B in the other panels. **e**, The IFT172-WD1 domain folded as a multimer with the CH domain of IFT57 forming a complex that is represented in the density of the IFT172 masked refinement map. **f**, The IFT54/20 (lime/pale green) bridge the gap in the IFT80-WD2 ring. **g**, Coloured density of Fig. 3d, showing our newly refined dynein average. Dynein repeats are alternating pink/purple, IFT-B2 is green. **h**, Side view of F. **i**, Same view as G, with density made translucent and the models docked in. **j**, The density in our new dynein average cropped out around the original dynein model (white) shows that the heavy chain undergoes a rearrangement in our newly refined model (purple), leaving an unmodelled density (inset). **k**, The unmodelled density likely corresponds to a Tctex1 dimer (green), linking the motor domains to the tail. **l**, A view of the top surface of IFT-B2, corresponding to the site where the dynein MTBD binds. **m**, The same view with surface charge representations shown, highlighting a positively charged patch where dynein binds.

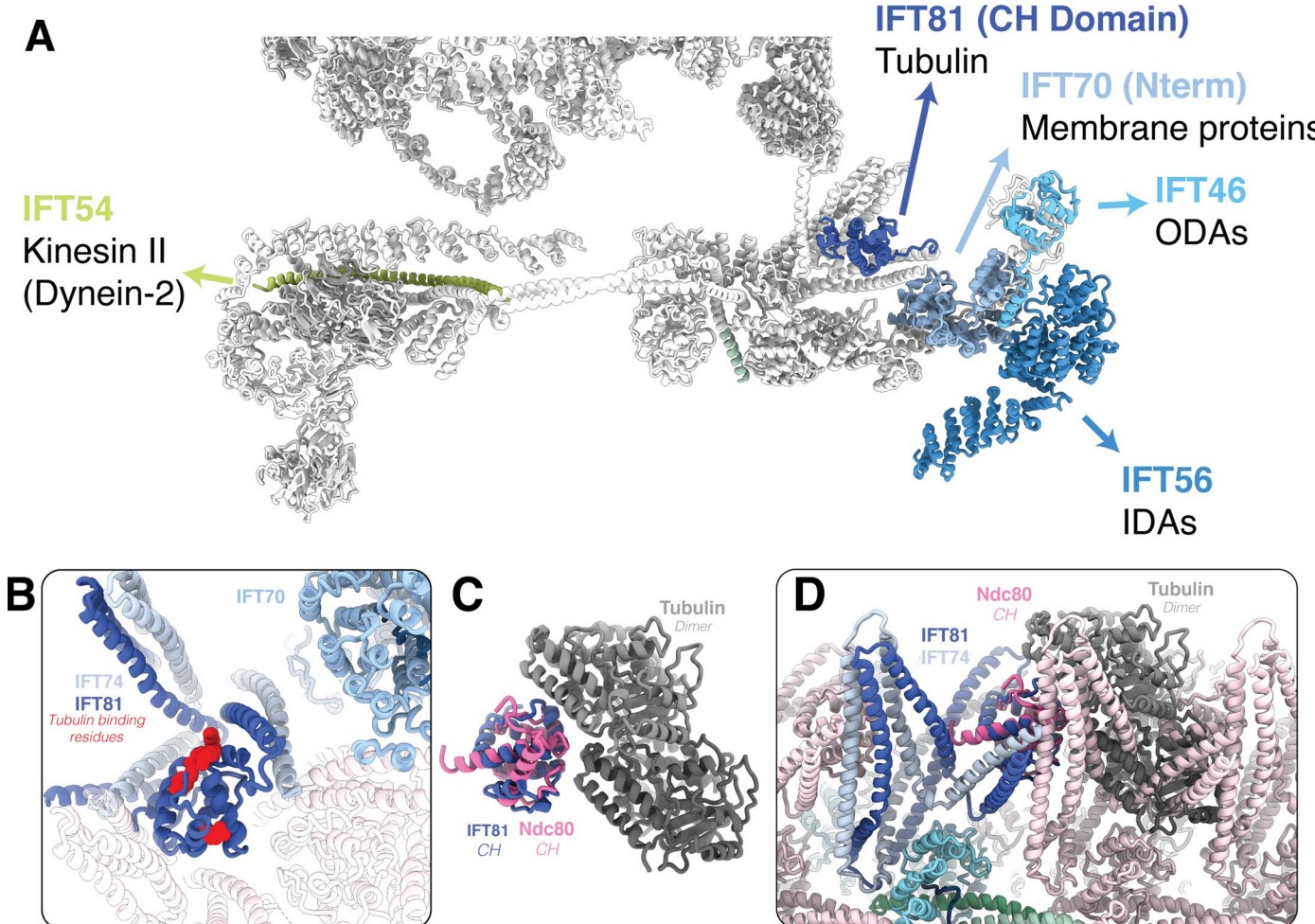

**Extended Data Fig. 8 | Cargo interactions in anterograde IFT trains. a**, The IFT-A and IFT-B models are displayed in grey, with regions of IFT-B previously linked biochemically to cargo transport labelled coloured. The large structural cargo interactions mostly occur at the edge of IFT-B1. IFT54 is thought to recruit kinesin II to anterograde trains, but this is not visible in our structure, probably due to flexibility. **b**, The CH domain of IFT81 (navy blue), with positive residues thought to be important for tubulin binding shown in red. Only a narrow space exists between the coiled coil domains of IFT81/74 nearby. **c**, Comparison between IFT81 CH domain (navy blue) and the CH domain of Ndc80 (pink) bound to microtubules (grey, PDB 3IZO), indicating strong structural homology between the two CH domains. **d**, The Ndc80:MT complex structure docked with the Ndc80-CH domain aligned to the IFT81-CH domain, simulating a potential interaction with tubulin cargo. Strong steric clashes occur between tubulin and IFT81/74 in the neighbouring repeat.

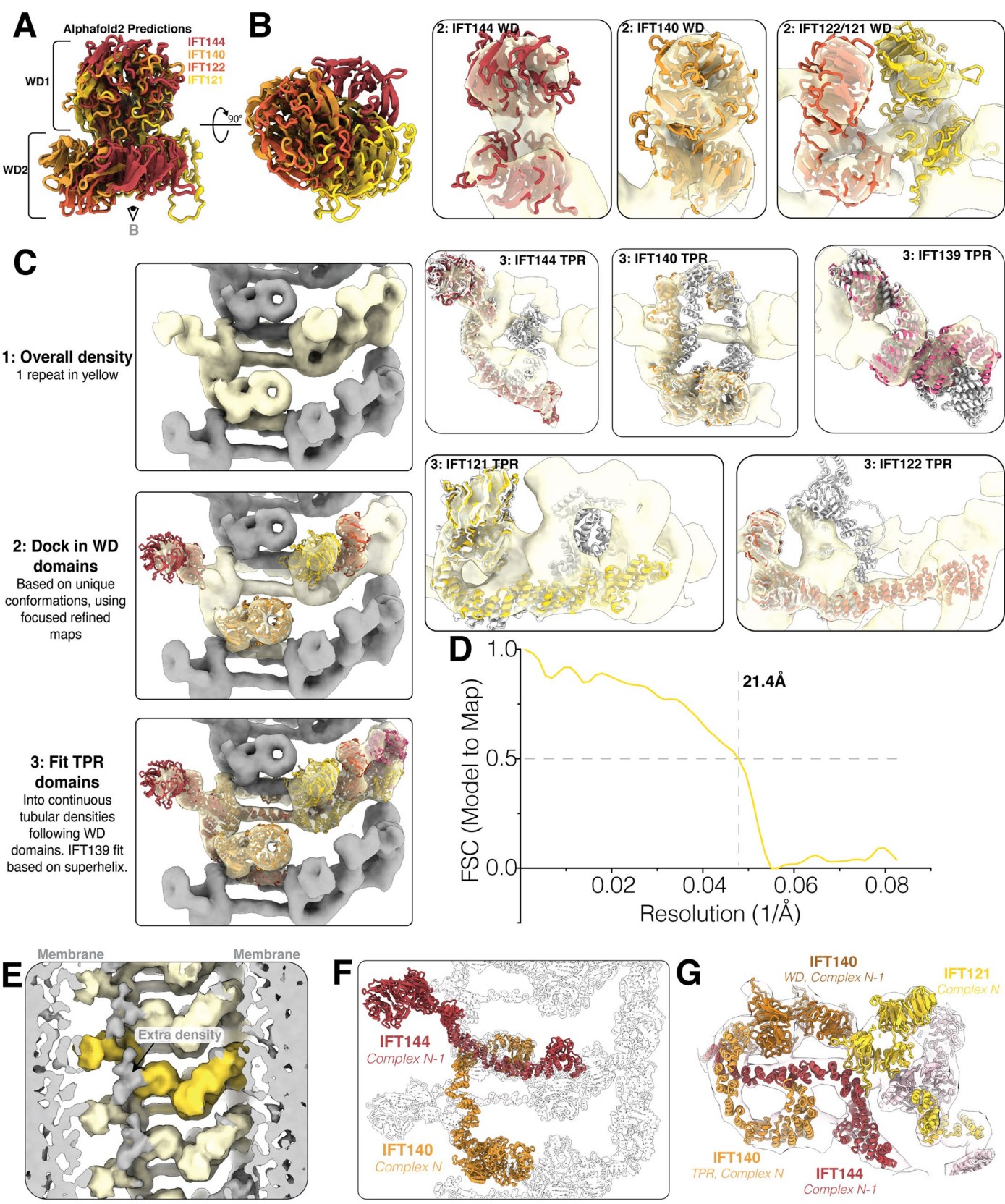

**Extended Data Fig. 9 | See next page for caption.**

**Extended Data Fig. 9 | The IFT-A polymer is built around four tandem WD domain proteins. a**, A comparison between the four tandem WD domains found in IFT-A, aligned at WD1. WD2 adopts a unique conformation relative to WD in each of the four proteins (with the TPR domain emerging at different places), allowing us to dock the models into the density. **b**, Equivalent to A, but with 90° rotation to provide a bottom view of the WD2 domains. **c**, A step-by-step guide of the model placements in IFT-A. *1:* One repeat of IFT-A highlighted in yellow. *2:* WD domains were docked into the density according to the angle between WD1 and WD2, and the exit of the TPR domain from WD2. Focused refinements were used for this positioning (as shown in inset panels for 2). *3:* TPR domains were fit into the continuous tubular densities emanating from each of the WD domains, with IFT139 identified as the remaining spiral density corresponding to the TPR superhelix. **d**, Model-to-map FSC curve for the IFTA model (into the overall 20.7 Å

3-repeat IFTA density). **e**, We lowpass filtered our IFT-A 3-repeat average, with regions containing part of our model coloured in yellow (dark yellow highlighting a single repeat). We see an extra density (grey) forming a bridge between the WD domains of IFT144 and IFT140 that is not formed by a protein in our model. **f**, Long distance interconnectivity between IFT144 and IFT140 from neighbouring complexes. The TPR domain of IFT140 (orange) reaches into the neighbouring complex and stabilize its copy of IFT144-TPR (dark red). **g**, Side view of F, with some extra subunits coloured and density shown. The TPR domain of IFT140 from the adjacent repeat stabilizes the conformation of IFT144. The WD domain of IFT140 (dark orange) sits on top of IFT144-TPR (both complex-1), meaning IFT140-TPR from complex 2 is determining the conformation of its neighbour. This stabilizes the binding site for IFT121-WD (yellow, complex 1).

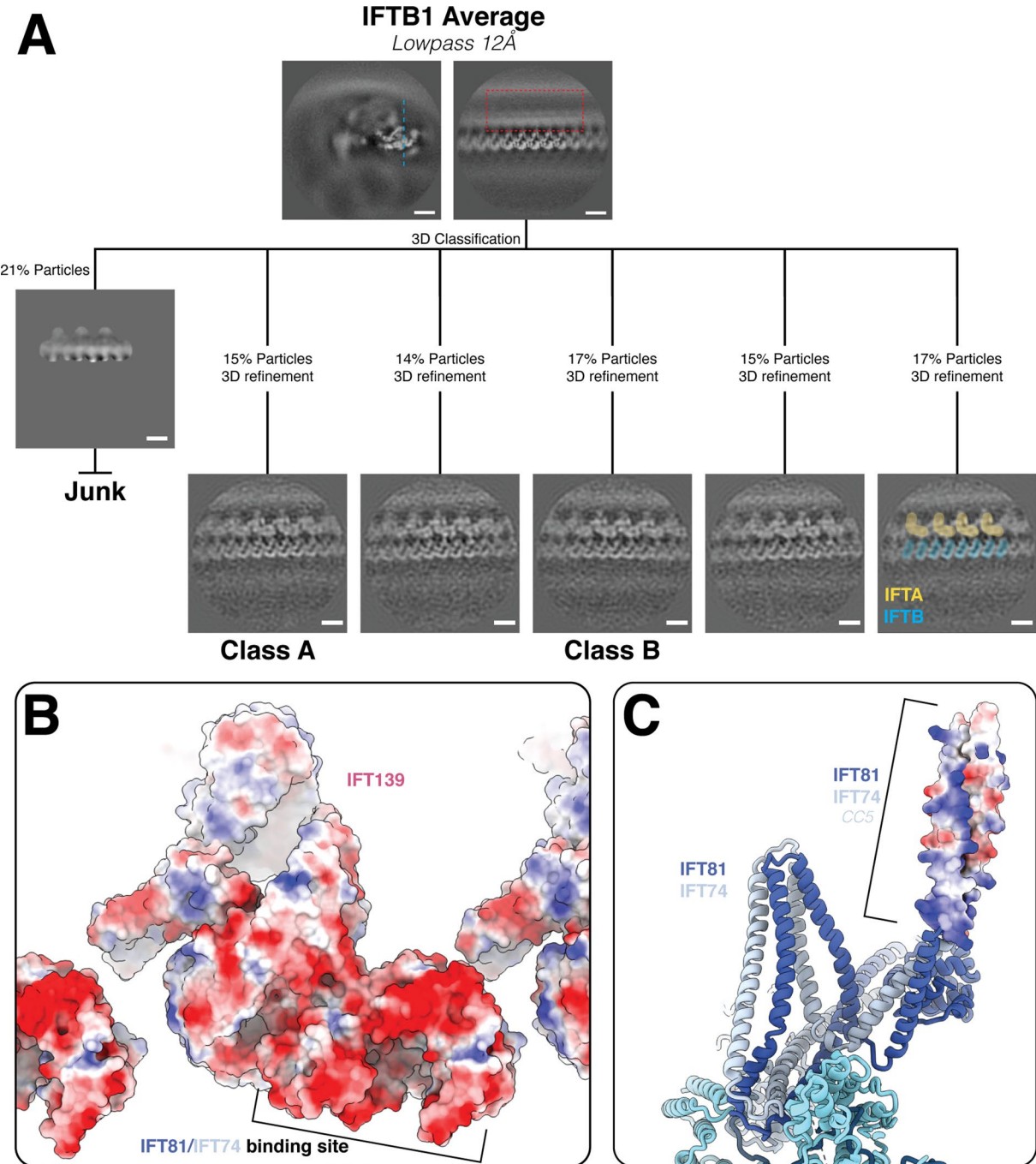

**Extended Data Fig. 10 | Classification of synchronous IFT-A and IFT-B averages. a**, Processing workflow of the classification of the IFT-B average to generate the classes in Fig. 5 that show synchronous IFT-A and IFT-B. Scale bars = 10 nm. **b**, Surface charge representation of IFT139 shows that the IFT81/74 binding site is strongly negatively charged. **c**, Surface charge representation of IFT81/74 CC5 shows that it is positively charged, facilitating its interaction with IFT139.

# Reporting Summary

## Statistics

For all statistical analyses, confirm that the following items are present in the figure legend, table legend, main text, or Methods section.

| n/a | Confirmed | |
|---|---|---|
| ☒ | ☐ | The exact sample size (*n*) for each experimental group/condition, given as a discrete number and unit of measurement |
| ☒ | ☐ | A statement on whether measurements were taken from distinct samples or whether the same sample was measured repeatedly |
| ☒ | ☐ | The statistical test(s) used AND whether they are one- or two-sided<br>*Only common tests should be described solely by name; describe more complex techniques in the Methods section.* |
| ☒ | ☐ | A description of all covariates tested |
| ☒ | ☐ | A description of any assumptions or corrections, such as tests of normality and adjustment for multiple comparisons |
| ☒ | ☐ | A full description of the statistical parameters including central tendency (e.g. means) or other basic estimates (e.g. regression coefficient) AND variation (e.g. standard deviation) or associated estimates of uncertainty (e.g. confidence intervals) |
| ☒ | ☐ | For null hypothesis testing, the test statistic (e.g. $F$, $t$, $r$) with confidence intervals, effect sizes, degrees of freedom and $P$ value noted<br>*Give P values as exact values whenever suitable.* |
| ☒ | ☐ | For Bayesian analysis, information on the choice of priors and Markov chain Monte Carlo settings |
| ☒ | ☐ | For hierarchical and complex designs, identification of the appropriate level for tests and full reporting of outcomes |
| ☒ | ☐ | Estimates of effect sizes (e.g. Cohen's *d*, Pearson's *r*), indicating how they were calculated |

*Our web collection on statistics for biologists contains articles on many of the points above.*

## Software and code

Policy information about availability of computer code

| | |
|---|---|
| Data collection | Cryo-electron tomograms were collected with Serial-EM V4.0 |
| Data analysis | Subtomogram averaging was performed with STOPGAP V0.7.1, Warp/M V1.0.9 and Relion V3.1.3. Structural prediction performed with Alphafold V2.1.1. MDFF performed on the NAMDinator web server (https://namdinator.au.dk/). |

For manuscripts utilizing custom algorithms or software that are central to the research but not yet described in published literature, software must be made available to editors and reviewers. We strongly encourage code deposition in a community repository (e.g. GitHub). See the Nature Portfolio guidelines for submitting code & software for further information.

## Data

Policy information about availability of data

All manuscripts must include a data availability statement. This statement should provide the following information, where applicable:
- Accession codes, unique identifiers, or web links for publicly available datasets
- A description of any restrictions on data availability
- For clinical datasets or third party data, please ensure that the statement adheres to our policy

The following maps have been deposited to the Electron microscopy data bank: IFT-B consensus of focused refinements (EMD-15977), IFT-B1 focused refinement (EMD-15978, IFT-B1 peripheral focused refinement as associated map), IFTB2 focused refinement (EMD-15979, IFT-B2 peripheral focused refinement as associated

map), IFT-B low-resolution overall map to validate consensus (EMD-16014) and IFTA (3-repeat map EMD-15980, 1 repeat and masked refinements as associated maps in this deposition). The IFT-B and IFT-A atomic models have been deposited to the protein data bank with the codes PDB-8BD7 and PDB-8BDA respectively.

## Human research participants

Policy information about studies involving human research participants and Sex and Gender in Research.

| Reporting on sex and gender | N/A |
|---|---|
| Population characteristics | N/A |
| Recruitment | N/A |
| Ethics oversight | N/A |

Note that full information on the approval of the study protocol must also be provided in the manuscript.

# Field-specific reporting

Please select the one below that is the best fit for your research. If you are not sure, read the appropriate sections before making your selection.

☒ Life sciences          ☐ Behavioural & social sciences          ☐ Ecological, evolutionary & environmental sciences

For a reference copy of the document with all sections, see nature.com/documents/nr-reporting-summary-flat.pdf

# Life sciences study design

All studies must disclose on these points even when the disclosure is negative.

| Sample size | We kept acquiring tomograms and adding subtomograms to our average until the overall resolution, as assessed by the gold-standard Fourier Shell Correlation (FSC) 0.143 criteria, stopped improving with the addition of more data. |
|---|---|
| Data exclusions | 3D Classification was performed in Relion to remove subtomograms that did not contribute to the overall average. |
| Replication | Cryo-EM grids used for data collection were frozen on 7 separate occasions (7 separate cultures = 7 biological replicates). |
| Randomization | Gold standard refinement includes randomisation into two groups and independent refinement in Relion. |
| Blinding | N/A because no variables/conditions were investigated in our study. |

# Reporting for specific materials, systems and methods

We require information from authors about some types of materials, experimental systems and methods used in many studies. Here, indicate whether each material, system or method listed is relevant to your study. If you are not sure if a list item applies to your research, read the appropriate section before selecting a response.

### Materials & experimental systems

| n/a | Involved in the study |
|---|---|
| ☒ | ☐ Antibodies |
| ☐ | ☒ Eukaryotic cell lines |
| ☒ | ☐ Palaeontology and archaeology |
| ☒ | ☐ Animals and other organisms |
| ☒ | ☐ Clinical data |
| ☒ | ☐ Dual use research of concern |

### Methods

| n/a | Involved in the study |
|---|---|
| ☒ | ☐ ChIP-seq |
| ☒ | ☐ Flow cytometry |
| ☒ | ☐ MRI-based neuroimaging |

## Eukaryotic cell lines

Policy information about cell lines and Sex and Gender in Research

| Cell line source(s) | Chlamydomonas reinhardtii wild-type (CC625) cells and CC625 cells with glycocalyx proteins FMG1A and FMG1B deleted by CRISPR |
|---|---|

| Authentication | PCR validation of cell lines. |
| --- | --- |
| Mycoplasma contamination | N/A |
| Commonly misidentified lines<br>(See ICLAC register) | N/A |

