## [Peer Review File · Nature Structural & Molecular Biology]

Peer Review Information

Manuscript Title: The Molecular Structure of IFT-A and IFT-B in Anterograde Intraflagellar Transport Trains

Corresponding author name(s): Dr. Gaia Pigino

Editorial Notes:

Reviewer Comments & Decisions:

Decision Letter, initial version:

Message: 27th Sep 2022

Dear Gaia,

Thank you again for submitting your manuscript "The Molecular Structure of IFT-A and IFT-B in Anterograde Intraflagellar Transport Trains". I apologize for the delay in responding, which resulted from the difficulty in obtaining suitable referee reports. Nevertheless, we now have comments (below) from the 2 reviewers who evaluated your paper. In light of those reports, we remain interested in your study and would like to see your response to the comments of the referees, in the form of a revised manuscript. Please be sure to address/respond to all concerns of the referees in full in a point-by-point response and highlight all changes in the revised manuscript text file. If you have comments that are intended for editors only, please include those in a separate cover letter.

We expect to see your revised manuscript within 6 weeks. If you cannot send it within this time, please contact us to discuss an extension; we would still consider your revision, provided that no similar work has been accepted for publication at NSMB or published elsewhere.

As you already know, we put great emphasis on ensuring that the methods and statistics reported in our papers are correct and accurate. As such, if there are any changes that

should be reported, please submit an updated version of the Reporting Summary along with your revision.

Reporting Summary:

When submitting the revised version of your manuscript, please pay close attention to our [Digital Image Integrity Guidelines](https://www.nature.com/nature-portfolio/editorial-policies/image-integrity).

Data availability: this journal strongly supports public availability of data. All data used in accepted papers should be available via a public data repository, or alternatively, as Supplementary Information. If data can only be shared on request, please explain why in your Data Availability Statement, and also in the correspondence with your editor. Please note that for some data types, deposition in a public repository is mandatory - more information on our data deposition policies and available repositories can be found below: <https://www.nature.com/nature-research/editorial-policies/reporting-standards#availability-of-data>

We require deposition of coordinates (and, in the case of crystal structures, structure factors) into the Protein Data Bank with the designation of immediate release upon publication (HPUB). Electron microscopy-derived density maps and coordinate data must be deposited in EMDB and released upon publication. Deposition and immediate release of NMR chemical shift assignments are highly encouraged. Deposition of deep sequencing and microarray data is mandatory, and the datasets must be released prior to or upon publication. To avoid delays in publication, dataset accession numbers must be supplied with the final accepted manuscript and appropriate release dates must be indicated at the

galley proof stage.

[Redacted]

Kind regards,
Florian

Dr Florian Ullrich
Associate Editor, Nature
Consulting Editor, Nature Structural & Molecular Biology
ORCID 0000-0002-1153-2040

Referee expertise:

Referee #1: cilia structure, cryo-ET

Referee #2: IFT function

Reviewers' Comments:

Reviewer #1:

Remarks to the Author:

This manuscript by Lacey et al. resolved the structure of intraflagellar transport (IFT) trains using an innovative combination of extensive cryo-electron tomography and structure predictions using alphafold 2. The author collected 600 tomograms and applied extensive subtomographic averaging to obtain the secondary structure of the IFT train. The resulting structure itself is a significant achievement in understanding the mechanism of intraflagellar transport, which is also related to many ciliopathies. Therefore, the reviewer highly recommends this paper to be published in Nature Structural and Molecular Biology with great enthusiasm.

However, the presentation of the results needs to be extensively revised. Currently, most of the main text is devoted to explaining the structural model of IFT, which is potential "an interpretation" by the authors. If this paper were about X-ray crystallography results, this is fine because extensive validation tools are available. Because this paper is so innovative in combining cryo-ET results and alphafold predictions, there are no established tools to validate the atomic model. Supplementary table 2 does indeed contain various validation scores, but these scores validate only "atomic" structures (approximately 2-3 angstrom resolution range) but do not validate larger domain structures, which are the main experimental results obtained by cryo-ET.

So, here are the major points that need to be addressed:

(1) The cryo-ET-derived map should be shown in more detail in Figure 1 because it is the main "result" of this paper.

(2) The process of the model building should be explained using figures (and possibly movies).

Although the modeling process (lines 491-534) is explained in the method section, it is hard for the reviewer to trace this process and make sure there is no other major alternative modeling. To help the reviewer and the reader of this paper, this modeling process should be graphically shown as supplementary figures or movies.

In addition, the reviewer would like to ask the authors to try to "validate" the resulting model. For example, the model vs. cryo-ET map FSC would be a good candidate, although such validations should be established by the cryo-EM community in the future.

(3) There is no description of data availability.

Because of the intricate structures of the IFT train and also the mandate of the NSMB journal, the authors should deposit both the cryo-ET map and the atomic model to the corresponding database and make them available. It is desirable that these data are also available during the reviewing process.

Minor comments:

(1) Figure 5K: the caption is missing.

(2) Line 506: all contained strong structural motifs that us to position -> that LET us to position

Reviewer #2:

Remarks to the Author:

In the present study, Lacey et al. have expanded on the previous pioneering work (Ref. 9)

from the same lab and proposed more detailed molecular models of IFT-B and IFT-A complexes in the *Chlamydomonas* anterograde IFT train. However, due to the resolution limitation of structural analysis based on cryo-ET, the models have been assembled with the aid of the AlphaFold2 predictions, and there are still unmodelled regions. Furthermore, the authors have not incorporated known crystal structures of IFT components into their models, although these crystal structures are partial.

Of course, I think such pioneering research deserves to be published to the world.

However, the authors have not performed any biochemical validation experiments on the models they have constructed or discussed the models in light of known biochemical data and crystal structures. Therefore, it is necessary to verify whether the molecular models including predictions using AlphaFold2 are indeed correct, in conjunction with biochemical data and crystal structures from other studies to date.

Major points:

1. There is uncertainty in the positioning of IFT81/74 in the model in Figure 2. The authors described (line 133) "The loop between IFT81/74 CC1 and CC2 forms the main attachment to the IFTB1 core by binding to the same cleft in IFT88 as IFT57/38 (Figure 2F/G).", although the C-terminal region of IFT81/74 (CC5-CC8) is not visible in this model. On the other hand, previous biochemical studies of *Chlamydomonas* IFT-B (Ref. 11, Taschner et al., 2014) and mammalian IFT-B (Ref. 23, Zhou et al., 2022) indicated that the C-terminal region of IFT81/74 is the attachment point for IFT52/46. Thus, the IFT-B1 model proposed by the authors is inconsistent with the biochemical data regarding not only the IFT-B1 binding region of IFT81/74 but also the IFT81/74-binding subunits of IFT-B1. In order to ensure the credibility of this IFT-B model, there must be a clear explanation regarding this inconsistency, including the validity of the AlphaFold2 predictions.
2. On the basis of the model in Figure 5, the authors described (line 348) "IFT139 has a strongly negatively charged surface and IFT81/74-CC5 is positively charged, making a favourable ionic interaction possible (Figure S10B/C). This interaction is also consistent with the mutations in IFT139 we find in this region (Figure 4D), which could affect IFT81/74 binding.", although IFT81/74-CC5 is not visible in the Figure 2 model. The interaction between IFT139 and IFT81/74-CC5 would not be possible without assuming that the positioning of IFT81/74 the Figure 2 model is correct. On the other hand, the Ref. 17 bioRxiv study, in which one of the authors of this paper is included as a co-author, found chemical cross-linking of IFT122 and IFT140 of IFT-A to IFT70, IFT88, and IFT172 of IFT-B, but did not mention that of IFT139 to IFT81/74. In addition, Kobayashi et al. (2021, *MBoC*, 32, 45) proposed on the basis of biochemical experiments that IFT122 and IFT144 of IFT-A and IFT88 and IFT52 of IFT-B mainly contribute to the interaction between IFT-A and IFT-B in the mammalian anterograde train. Therefore, the models in Figures 2 and 5 need to be properly discussed in relation to the data in these two papers, including the validity of predictions with AlphaFold2. In addition, the Ref. 13 study (Wachter et al., 2019, *EMBOJ*) revealed the crystal structure of the CC1-CC6 of *Trypanosoma* IFT81/74 in complex with GTP-bound IFT22/RabL5. This crystal structure should be incorporated into the authors' model if possible and discussed with respect to their claims.
3. For the IFT-B2 model (Figure 3A-C), the authors described (line 145) "The second WD domain of both these proteins (IFT172 and IFT80) forms an incomplete circle (Figure 3A-C, Figure S7F), particularly dramatically in the case of IFT172." and (line 166) "IFT80-TPR wraps around the N-terminal TPR motifs of IFT172 from the neighbouring repeat." On the other hand, previous biochemical studies (Ref. 10, Taschner et al., 2016, *EMBOJ*; and Katoh et al, 2016, *JBC*, 291, 10962) suggested that IFT172 and IFT80 interact with each other via their C-terminal regions and that the WD domains of IFT172 and IFT80 interact

with IFT57 and IFT38, respectively, via their N-terminal CH domains, the former of which is not visible in the Figure 3 model. Refer to these papers and discuss the IFT-B2 model. 4. Most of DISCUSSION is devoted to the retrograde train and train remodeling from anterograde to retrograde. However, this study did not analyze the retrograde train. DISCUSSION therefore needs to be restructured to focus mainly on the validation of the molecular models for the anterograde train and other issues, while also leaving some discussion about train remodeling.

Minor points:

1. The title is incorrect: IFTA-A → IFT-A (this is an important point)

2. Hyphens that should link elements of compound words are frequently missing.

e.g., IFTA → IFT-A; IFTB → IFT-B; 11.5nm repeats → 11.5-nm repeats

Spaces are frequently missing between numbers and units.

e.g., ~30Å resolution → ~30 Å resolution

In the Ref. 9 paper (Jordan et al., 2018, NCB), the authors were able to describe these points exactly. However, these points are not addressed correctly in this manuscript.

3. Typo

Line 117: IFT58/37 → IFT57/38

Line 254: (Figure L/M) → (Figure S7L/M)

uL, um → μL, μm

Author Rebuttal to Initial comments

Reviewer 1 Major Points

(1) The cryo-ET-derived map should be shown in more detail in Figure 1 because it is the main "result" of this paper.

We agree with the comment and have updated Figure 1 to better allow readers to evaluate the quality of the map and the fit of the models within it. Panels 1C and 1E are top views of the IFT-B and IFT-A density respectively, with one repeat highlighted with colour. Panels 1D/F are the same views with the central coloured repeat now partially transparent and the molecular model we built docked in. This shows the quality of the density and the fit of the model into the density much better than in the original manuscript.

(2) The process of the model building should be explained using figures (and possibly movies).

Although the modeling process (lines 491-534) is explained in the method section, it is hard for the reviewer to trace this process and make sure there is no other major alternative modeling. To help the reviewer and the reader of this paper, this modeling process should be graphically shown as supplementary figures or movies.

In addition, the reviewer would like to ask the authors to try to "validate" the resulting model. For example, the model vs. cryo-ET map FSC would be a good candidate, although such validations should be established by the cryo-EM community in the future.

To address this point, we now include a graphical step-by-step workflow of our fitting process for IFT-B and IFT-A in Supplementary figures 5 and 9 respectively. We show how we started by docking in proteins or domains where there is an unambiguous fit of the original AlphaFold2 prediction into the density. We then show how the remaining models were incorporated into the remaining density, based on structural features and previously characterized interactions. In these step, we show comparisons of the original AlphaFold2 models with our final model, and the movements necessary to fit the models into the density. We hope that this clearly illustrates the decisions we made during our model fitting process, and reinforces confidence in our final model.

We have also included updated supplementary movies to better show the quality of the fit of the model inside the density.

We have made one modification to our model compared to the original submission. This is in response to a preprint containing a single particle structure of isolated IFT-A complexes released after our initial submission ¹. In our density, IFT-A complexes are continuously interconnected, making the definition of a single IFT-A complex arbitrary. The single particle structure shows that in our original assignation IFT144 and IFT140 belong to the adjacent complex. We have therefore updated our model to match this repeating unit definition, as shown in an updated Figure 4. The conformations/structures of IFT144/140 remain unchanged, as do their position in the IFT-A polymer. Our structure is otherwise remarkably similar to the single particle structure, and the interpretation of our model remains is mostly unchanged, with IFT144 and IFT140 both still extending into the adjacent complexes. However, now we see that the connection between IFT-A and IFT-B that we assign to the IFT172 C-terminus is actually bridging two non-consecutive IFT-A complexes (i.e. complex N and N+2). This further suggests that IFT172 helps guide IFT-A polymerization by establishing longer-range lateral interactions in the IFTA polymer. We have added a sentence to describe this observation to lines 358-360 of the results section, and updated the colouring in Figure 5G to highlight the long-range interaction .

(3) There is no description of data availability.

Because of the intricate structures of the IFT train and also the mandate of the NSMB journal, the authors should deposit both the cryo-ET map and the atomic model to the corresponding database and make them available. It is desirable that these data are also available during the reviewing process.

Data has been deposited to the EMDB and PDB, with accession codes provided in the "Data availability" section of the manuscript. We have uploaded these files in a zip folder with our resubmission for the reviewers to view before they are released on the public databases.

Reviewer 1 Minor Points

(1) Figure 5K: the caption is missing.

(2) Line 506: all contained strong structural motifs that us to position -> that LET us to position

Both points have been corrected in the revised manuscript.

Reviewer 2 Major Points

1. There is uncertainty in the positioning of IFT81/74 in the model in Figure 2. The authors described (line 133) “The loop between IFT81/74 CC1 and CC2 forms the main attachment to the IFTB1 core by binding to the same cleft in IFT88 as IFT57/38 (Figure 2F/G).”, although the C-terminal region of IFT81/74 (CC5-CC8) is not visible in this model. On the other hand, previous biochemical studies of *Chlamydomonas* IFT-B (Ref. 11, Taschner et al., 2014) and mammalian IFT-B (Ref. 23, Zhou et al., 2022) indicated that the C-terminal region of IFT81/74 is the attachment point for IFT52/46. Thus, the IFT-B1 model proposed by the authors is inconsistent with the biochemical data regarding not only the IFT-B1 binding region of IFT81/74 but also the IFT81/74-binding subunits of IFT-B1. In order to ensure the credibility of this IFT-B model, there must be a clear explanation regarding this inconsistency, including the validity of the AlphaFold2 predictions.

Regarding points 1-3 of reviewer 2, we agree that we did not sufficiently address differences between the interactions seen in our model and previously mapped interactions in our initial submission. To address this we have updated the discussion section, with the changes outlined below.

In general, we propose that differences between our model and previous data can be explained by the context in which they are observed. We imaged fully assembled anterograde trains, however these only represent part of the “life cycle” of IFT proteins. We have previously shown that retrograde trains adopt a different (although currently unknown) conformation, and different conformations are likely to occur in individual (sub)complexes before assembly. The previously mapped interactions are based on recombinant or native samples that have undergone some form of purification (e.g. co-immunoprecipitation, size-exclusion chromatography) taking them out of their native environment. We suggest that the interactions observed in these studies are those seen in isolated complexes prior to polymerization. This is supported by the low levels of oligomerization seen after purification^(1,2).

In the case of IFT81/74, this outlook is supported by a subsequent preprint published by the Lorentzen lab³. They used cross-linking mass spectrometry to show that IFT81/74 can bind to two mutually exclusive sites in IFTB1. The “main” interaction is the previously established interaction between IFT81/74 C-terminus and IFT52/46, but the second is the interaction with IFT88 and IFT70 that we observe. We therefore now propose that stabilization of IFT81/74 at the IFT88/70 site is a step in the polymerization of anterograde trains. This argument is incorporated into our amended discussion in lines 398 to 404.

2. On the basis of the model in Figure 5, the authors described (line 348) “IFT139 has a strongly negatively charged surface and IFT81/74-CC5 is positively charged, making a favourable ionic interaction possible (Figure S10B/C). This interaction is also consistent with the mutations in IFT139 we find in this region (Figure 4D), which could affect IFT81/74 binding.”, although IFT81/74-CC5 is not visible in the Figure 2 model. The interaction between IFT139 and IFT81/74-CC5 would not be possible without assuming that the positioning of IFT81/74 the Figure 2 model is correct. On the other hand, the Ref. 17 bioRxiv study, in which one of the authors of this paper is included as a co-author, found chemical cross-linking of IFT122 and IFT140 of IFT-A to IFT70, IFT88, and IFT172 of IFT-B, but did not

mention that of IFT139 to IFT81/74. In addition, Kobayashi et al. (2021, MBoC, 32, 45) proposed on the basis of biochemical experiments that IFT122 and IFT144 of IFT-A and IFT88 and IFT52 of IFT-B mainly contribute to the interaction between IFT-A and IFT-B in the mammalian anterograde train. Therefore, the models in Figures 2 and 5 need to be properly discussed in relation to the data in these two papers, including the validity of predictions with AlphaFold2. In addition, the Ref. 13 study (Wachter et al., 2019, EMBOJ) revealed the crystal structure of the CC1-CC6 of Trypanosoma IFT81/74 in complex with GTP-bound IFT22/RabL5. This crystal structure should be incorporated into the authors' model if possible and discussed with respect to their claims.

These differences can also be explained by the different experimental approaches. In Ref 17 (McCafferty et al, 2022), an extensive purification regime was used prior to cross-linking, meaning that cross links are unlikely to represent the native polymerized anterograde train conformation. As is the case above, we are not disagreeing with the previous data, rather suggesting that it represents a different conformation to the state we imaged.

The IFT88:IFT144/122 interaction is directly compatible with our model. The IFT88 C-terminus is predicted to be long and disordered, and was not included in our overall model. However, it's location would easily allow it to contact these IFTA subunits. However, this length and disorder means that it is unlikely to induce the formation of the tightly coupled IFTA and IFTB polymer conformation we observe in our density. As such, we propose that it could be an interaction used to form an initial attachment, with the two ordered interactions we observe in our density being used to stabilize the anterograde conformation of IFTA relative to IFTB. This argument is incorporated into our amended discussion in lines 421 to 428.

Regarding the crystal structure of IFT81/74 – we chose to use the Alphafold2 model since the crystal structure used the protein from a different species (24% sequence identity between Clamydomonas and Trypanosoma in both). Furthermore, the IFT81/74 crystal structure is still only a fragment, and when we started model building it was desirable to have full-length models to better determine how the models fit into the density. However, the crystal structure and the Alphafold2 structure were very similar (as far all cases where a crystal structure had been solved), and ultimately would not have resulted in a different final model. This decision is described in the methods section lines 545 to 549.

3. For the IFT-B2 model (Figure 3A-C), the authors described (line 145) “The second WD domain of both these proteins (IFT172 and IFT80) forms an incomplete circle (Figure 3A-C, Figure S7F), particularly dramatically in the case of IFT172.” and (line 166) “IFT80-TPR wraps around the N-terminal TPR motifs of IFT172 from the neighbouring repeat.” On the other hand, previous biochemical studies (Ref. 10, Taschner et al., 2016, EMBOJ; and Katoh et al, 2016, JBC, 291, 10962) suggested that IFT172 and IFT80 interact with each other via their C-terminal regions and that the WD domains of IFT172 and IFT80 interact with IFT57 and IFT38, respectively, via their N-terminal CH domains, the former of which is not visible in the Figure 3 model. Refer to these papers and discuss the IFT-B2 model.

The interactions between IFT172 and IFT80 that involve the WD domains are specifically at the lateral interface between repeating units, again suggesting that they would likely not be present in the conditions used for pull-downs. We discuss this point in lines 406 to 419.

4. Most of DISCUSSION is devoted to the retrograde train and train remodeling from anterograde to retrograde. However, this study did not analyze the retrograde train. DISCUSSION therefore needs to be restructured to focus mainly on the validation of the molecular models for the anterograde train and other issues, while also leaving some discussion about train remodeling.

We have significantly updated the discussion in the revised manuscript. These changes address the reviewers concerns regarding the biochemical validation of previously observed interactions and the use of AF2 predictions to build the model.

Reviewer 2 Minor Points

1. The title is incorrect: IFTA-A → IFT-A (this is an important point)

2. Hyphens that should link elements of compound words are frequently missing.

e.g., IFTA → IFT-A; IFTB → IFT-B; 11.5nm repeats → 11.5-nm repeats

Spaces are frequently missing between numbers and units.

e.g., ~30Å resolution → ~30 Å resolution

In the Ref. 9 paper (Jordan et al., 2018, NCB), the authors were able to describe these points exactly. However, these points are not addressed correctly in this manuscript.

3. Typo

Line 117: IFT58/37 → IFT57/38

Line 254: (Figure L/M) → (Figure S7L/M)

uL, um → μL, μm

All the points raised here have now been corrected.

Citations

1. Hesketh, S. J., Mukhopadhyay, A. G., Nakamura, D., Toropova, K. & Roberts, A. J. IFT-A Structure Reveals Carriages for Membrane Protein Transport into Cilia. *bioRxiv* 2022.08.09.503213 (2022) doi:10.1101/2022.08.09.503213.

2. Taschner, M., Kotsis, F., Braeuer, P., Kuehn, E. W. & Lorentzen, E. Crystal structures of IFT70/52 and IFT52/46 provide insight into intraflagellar transport B core complex assembly. *J. Cell Biol.* **207**, 269–282 (2014).

3. Petriman, N. A. *et al.* Biochemically validated structural model of the 15-subunit IFT-B complex. <http://biorxiv.org/lookup/doi/10.1101/2022.08.20.504624> (2022) doi:10.1101/2022.08.20.504624.

Decision Letter, first revision:

Message: Our ref: NSMB-A46745A

21st Nov 2022

Dear Gaia,

Thank you for submitting your revised manuscript "The Molecular Structure of IFT-A and IFT-B in Anterograde Intraflagellar Transport Trains" (NSMB-A46745A). It has now been seen by two of the original referees and their comments are below. The reviewers find that the paper has improved in revision, and therefore we'll be happy in principle to publish it in Nature Structural & Molecular Biology, pending minor revisions to satisfy the referees' final requests and to comply with our editorial and formatting guidelines.

To facilitate our work at this stage, we would appreciate if you could send us the main text as a word file. Please make sure to copy the NSMB account (cc'ed above).

Kind regards,
Florian

Dr Florian Ullrich
Associate Editor, Nature
Consulting Editor, Nature Structural & Molecular Biology
ORCID 0000-0002-1153-2040

Reviewer #1 (Remarks to the Author):

I think the manuscript was properly revised, especially the process of model building was described in detail and the accuracy of the model is (somewhat) tested by comparing the model vs the cryo-EM map. Therefore the manuscript can be accepted.

Reviewer #2 (Remarks to the Author):

The revised manuscript has been improved almost satisfactorily. However, there are several errors, in particular, in the newly added part of DISCUSSION.

1. Correct the numbering of the reference papers in DISCUSSION.

Lines 387, 388, 399, 402, 406, 411, 415 & 423

2. IFTA, IFTB, IFTB1, IFTB2 -> IFT-A, IFT-B, IFT-B1, IFT-B2 in DISCUSSION.

3. Line 444, (Figure S5B) ?; Line 583, (Figure S5A) ?

4. Reference 42, bioRxiv 2021 -> Curr. Biol. 2022

Decision Letter, author guidance:

Message: Our ref: NSMB-A46745A

28th Nov 2022

Dear Dr. Pigino,

Thank you for your patience as we've prepared the guidelines for final submission of your Nature Structural & Molecular Biology manuscript, "The Molecular Structure of IFT-A and IFT-B in Anterograde Intraflagellar Transport Trains" (NSMB-A46745A). Please carefully follow the step-by-step instructions provided in the attached file, and add a response in each row of the table to indicate the changes that you have made. Please also check and comment on any additional marked-up edits we have proposed within the text. Ensuring that each point is addressed will help to ensure that your revised manuscript can be swiftly handed over to our production team.

In recognition of the time and expertise our reviewers provide to Nature Structural & Molecular Biology's editorial process, we would like to formally acknowledge their contribution to the external peer review of your manuscript entitled "The Molecular Structure of IFT-A and IFT-B in Anterograde Intraflagellar Transport Trains". For those reviewers who give their assent, we will be publishing their names alongside the published article.

Nature Structural & Molecular Biology offers a Transparent Peer Review option for new original research manuscripts submitted after December 1st, 2019. As part of this initiative, we encourage our authors to support increased transparency into the peer review process by agreeing to have the reviewer comments, author rebuttal letters, and editorial decision letters published as a Supplementary item. When you submit your final files please clearly state in your cover letter whether or not you would like to participate in this initiative. Please note that failure to state your preference will result in delays in

accepting your manuscript for publication.

Cover suggestions

As you prepare your final files we encourage you to consider whether you have any images or illustrations that may be appropriate for use on the cover of Nature Structural & Molecular Biology.

Nature Structural & Molecular Biology has now transitioned to a unified Rights Collection system which will allow our Author Services team to quickly and easily collect the rights and permissions required to publish your work. Approximately 10 days after your paper is formally accepted, you will receive an email in providing you with a link to complete the grant of rights. If your paper is eligible for Open Access, our Author Services team will also be in touch regarding any additional information that may be required to arrange payment for your article.

Please note that *Nature Structural & Molecular Biology* is a Transformative Journal (TJ). Authors may publish their research with us through the traditional subscription access route or make their paper immediately open access through payment of an article-processing charge (APC). Authors will not be required to make a final decision about access to their article until it has been accepted. [Find out more about Transformative Journals](https://www.springernature.com/gp/open-research/transformative-journals)

Authors may need to take specific actions to achieve [compliance](https://www.springernature.com/gp/open-research/funding/policy-compliance-faqs) with funder and institutional open access mandates. If your research is supported by a funder that requires immediate open access (e.g. according to [Plan S principles](https://www.springernature.com/gp/open-research/plan-s-compliance)) then you should select the gold OA route, and we will direct you to the compliant route where possible. For authors selecting the subscription publication route, the journal's standard licensing terms will need to be accepted, including [self-archiving policies](https://www.nature.com/nature-portfolio/editorial-policies/self-archiving-and-license-to-publish). Those licensing terms will supersede any other terms that the author or any third party may assert apply to any

version of the manuscript.

Please use the following link for uploading these materials:
[Redacted]

Best regards,

Aimee Frier
Editorial Assistant
Nature Structural & Molecular Biology
nsmb@us.nature.com

On behalf of

Florian Ullrich, Ph.D.
Associate Editor
Nature Structural & Molecular Biology
ORCID 0000-0002-1153-2040

Reviewer #1:
None

Reviewer #2:
Remarks to the Author:
The revised manuscript has been improved almost satisfactorily.
However, there are several errors, in particular, in the newly added part of DISCUSSION.

1. Correct the numbering of the reference papers in DISCUSSION.
Lines 387, 388, 399, 402, 406, 411, 415 & 423
2. IFTA, IFTB, IFTB1, IFTB2 -> IFT-A, IFT-B, IFT-B1, IFT-B2 in DISCUSSION.
3. Line 444, (Figure S5B) ?; Line 583, (Figure S5A) ?
4. Reference 42, bioRxiv 2021 -> Curr. Biol. 2022

Author Rebuttal, first revision:

We have now addressed all these points in our revised manuscript.

Final Decision Letter:

Message 1st Dec 2022

:

Dear Gaia,

We are now happy to accept your revised paper "The Molecular Structure of IFT-A and IFT-B in Anterograde Intraflagellar Transport Trains" for publication as a Article in Nature Structural & Molecular Biology.

As soon as your article is published, you can generate your shareable link by entering the DOI of your article here: http://authors.springernature.com/share. Corresponding authors will also receive an automated email with the shareable link

Note the policy of the journal on data deposition:

<http://www.nature.com/authors/policies/availability.html>.

Your paper will be published online soon after we receive proof corrections and will appear in print in the next available issue. You can find out your date of online publication by contacting the production team shortly after sending your proof corrections. Content is published online weekly on Mondays and Thursdays, and the embargo is set at 16:00 London time (GMT)/11:00 am US Eastern time (EST) on the day of publication. Now is the time to inform your Public Relations or Press Office about your paper, as they might be interested in promoting its publication. This will allow them time to prepare an accurate and satisfactory press release. Include your manuscript tracking number (NSMB-A46745B) and our journal name, which they will need when they contact our press office.

About one week before your paper is published online, we shall be distributing a press release to news organizations worldwide, which may very well include details of your work. We are happy for your institution or funding agency to prepare its own press release, but it must mention the embargo date and Nature Structural & Molecular Biology. If you or your Press Office have any enquiries in the meantime, please contact press@nature.com.

Please note that *Nature Structural & Molecular Biology* is a Transformative Journal (TJ). Authors may publish their research with us through the traditional subscription access route or make their paper immediately open access through payment of an article-processing charge (APC). Authors will not be required to make a final decision about access to their article until it has been accepted. [Find out more about Transformative Journals](https://www.springernature.com/gp/open-research/transformative-journals)

Authors may need to take specific actions to achieve [compliance](https://www.springernature.com/gp/open-research/funding/policy-compliance-faqs) with funder and institutional open access

mandates. If your research is supported by a funder that requires immediate open access (e.g. according to [Plan S principles](https://www.springernature.com/gp/open-research/plan-s-compliance)) then you should select the gold OA route, and we will direct you to the compliant route where possible. For authors selecting the subscription publication route, the journal's standard licensing terms will need to be accepted, including [self-archiving policies](https://www.springernature.com/gp/open-research/policies/journal-policies). Those licensing terms will supersede any other terms that the author or any third party may assert apply to any version of the manuscript.

Kind regards,
Florian

Dr Florian Ullrich
Associate Editor, Nature
Consulting Editor, Nature Structural & Molecular Biology
ORCID 0000-0002-1153-2040
